# TEMPTABQA: Temporal Question Answering for Semi-Structured Tables

**Vivek Gupta**[1‡*] , **Pranshu Kandoi**[2†], **Mahek Bhavesh Vora**[2†], **Shuo Zhang**[3‡]
**Yujie He**[3], **Ridho Reinanda**[3], **Vivek Srikumar**[4]
[1]University of Pennsylvania, [2]IIT Guwahati, [3]Bloomberg, [4]University of Utah,
gvivek@cis.upenn.edu, {k.pranshu,v.mahek}@iitg.ac.in, svivek@cs.utah.edu,
{szhang611, yhe247, rreinanda}@bloomberg.net

## Abstract

Semi-structured data, such as Infobox tables, often include temporal information about entities, either implicitly or explicitly. *Can current NLP systems reason about such information in semi-structured tables?* To tackle this question, we introduce the task of temporal question answering on semi-structured tables. We present a dataset, TEMPTABQA, which comprises 11,454 question-answer pairs extracted from 1,208 Wikipedia Infobox tables spanning more than 90 distinct domains. Using this dataset, we evaluate several state-of-the-art models for temporal reasoning. We observe that even the top-performing LLMs lag behind human performance by more than 13.5 F1 points. Given these results, our dataset has the potential to serve as a challenging benchmark to improve the temporal reasoning capabilities of NLP models.

## 1 Introduction

Reasoning about temporal aspects of factual information presents a fundamental challenge for contemporary Natural Language Processing (NLP) systems. Factual information related to an entity often evolves over time, and understanding it requires understanding the scope of knowledge and temporal intervals. Furthermore, this factual information is also scattered across semi-structured data in several different forms. These forms include both implicit and explicit representations (see Figure 1 for an example). The wide prevalence of these characteristics creates major challenges for NLP models. It requires these models to effectively handle changes over time and extract valuable insights from time-dependent data.

Previous studies have primarily concentrated on question answering (Pasupat and Liang, 2015; Krishnamurthy et al., 2017) and inference (Gupta

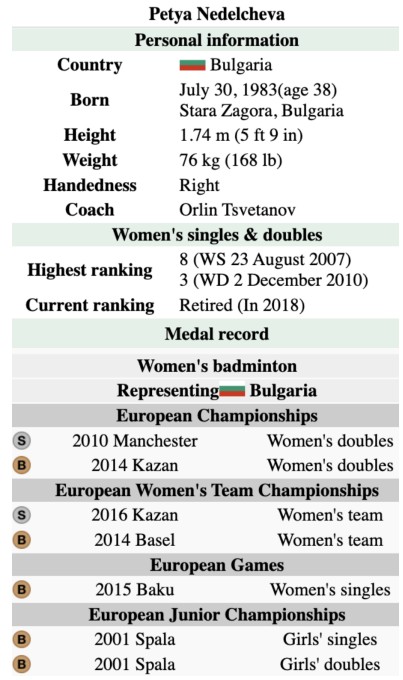

Q1: How many years elapsed between Nedelcheva's $1^{st}$ and last silver medals? A1: 6 years
Q2: How many bronze medals has Nedelcheva's won since 2013? A2: 3
Q3: Which country did Nedelcheva's represent in the 2015 European Games? A3: Bulgaria
Q4: What was the ranking difference between Nedelcheva's top doubles and singles performances? A4: 5

Figure 1: A semi-structured table of women badminton players (source: Wikipedia) along with accompanying temporal questions and their respective answers form TEMPTABQA.

et al., 2020; Chen et al., 2020b) concerning numerical aspects of semi-structured tables. They typically examine tables without time-related information and focus on queries with predominantly non-temporal contexts. Research on temporal aspects in entity-centric tables, such as Wikipedia Infoboxes, has been limited (Gupta et al., 2020; Neeraja et al., 2021; Kumar et al., 2022). Morales et al. (2016) introduced question answering in an entity-centric table context. However, the questions in the dataset are simple and non-temporal, and it's worth noting that the dataset is not open source. Existing studies have only considered a few temporal aspects and addressed a small number of

---
*Work done during an internship at Bloomberg. †Equal contributions. ‡Corresponding authors.

time-related factors. Advances in modeling techniques, including table pre-training and targeted fine-tuning, have substantially improved reasoning on semi-structured tables (Müller et al., 2021; Eisenschlos et al., 2020). Moreover, large language models (LLMs) have exhibited impressive performance across various domains, such as general, finance, and medical, demonstrating their mathematical, knowledge-based, and common-sense reasoning capabilities (Chen et al., 2021d; Aly et al., 2021; Wang et al., 2021; Lu et al., 2023). However, the effectiveness of these models in handling temporal aspects remains understudied. Consequently, this paper seeks to address the following research question: *"Can modern NLP models effectively reason about temporal information in semi-structured tables?"*

To effectively address the above question, we introduce a new task called *temporal question answering on entity-centric semi-structured tables*. Figure 1 shows an example. We curate a comprehensive (covering diverse domains), specialized (temporally aligned), and human-verified dataset, TEMPTABQA. It consists of 11,454 question-answer pairs extracted from 1,208 Wikipedia Infobox tables across more than 90 domains. Both the tables and questions in TEMPTABQA are encompass numerous temporal terms. For example, the Figure 1 table involve multiple dates, age, years, and the corresponding questions also incorporate temporal terms, such as first and last, years (since 2013), ranking, and more. These enhanced tables incorporate temporal information, ensuring that all queries have time-related components. This dataset is the first to explore question-answering and temporal reasoning in semi-structured data.

We conduct analysis of temporal reasoning challenges in TEMPTABQA, offering both qualitative and quantitative insights. Most questions in TEMPTABQA are abstractive and necessitate mathematical calculations over temporal concepts to arrive at correct answers. The dataset also encompasses additional test sets to evaluate reasoning in rare domains. Our findings indicate that temporal reasoning in TEMPTABQA poses greater challenges compared to non-temporal reasoning in previous tabular datasets. Our assessment of contemporary NLP systems on the TEMPTABQA benchmark exposes their subpar performance in comparison to humans. Humans excel at temporal reasoning, delivering accurate answers and in-depth explanations. In contrast, our error analysis shows that models frequently make mistakes, particularly when faced with complex temporal reasoning questions. Consequently, our dataset serves as a challenging testbed for investigating effective temporal reasoning within semi-structured information.

Our paper marks a significant milestone by pioneering the creation of complex temporal question answering datasets, specifically tailored to entity-centric tables. Our primary objective was to introduce a novel challenge – addressing intricate temporal questions within this context. The TEMPTABQA dataset not only demands sophisticated reasoning but also necessitates a firm grasp of temporal common sense, adept handling of arithmetic and numerical aspects. Our work sheds light on the unique temporal specificity of this dataset, setting it apart from existing models. Furthermore, we delves deep into this differentiation, offering a comprehensive array of statistics and analyses that illuminate the multitude of temporal reasoning challenges posed by the dataset. The findings from above enhance our understanding of temporal reasoning in tables and encourage further research.

The TEMPTABQA dataset can be accessed at `https://zenodo.org/records/10022927`. For relevant analysis and modeling scripts refer at `https://temptabqa.github.io`.

## 2 Motivation

**Dynamic Nature of Information.** Tables serve as structured representations that organize and record diverse information types, making them highly useful for studying an entity's timeline. They offer a comprehensive record of events, enabling clear visualization of the evolution of various aspects over time. By capturing a chronological sequence of events, tables allow us to analyze the progression of positions, changes in marital status, and the acquisition of awards, serving as reliable sources for temporal reasoning. Additionally, entity-centric tables, such as Wikipedia Infoboxes, significantly differ from both unstructured and fully structured data (SQL tables and KGs). These tables possess a semi-structured nature and store data in intricate implicit forms, as discussed in the context of semi-structured tables (Gupta et al., 2020).

**Tables in the Real World.** Complex temporal question answering applied to entity-centric semi-structured tables, like Wikipedia Infoboxes, has

broad applicability across various fields. In historical research and education, it helps scholars, historians, and students extract precise historical details, while in financial analysis, it empowers analysts with historical financial data for informed investment decisions. In medical research, it aids in accessing historical medical data and clinical trial timelines, and legal professionals use it to review historical legal records. Journalists gain historical context, linguists analyze language dynamics, and businesses optimize supply chains through historical data. Environmental researchers, policy analysts, travelers, software developers, and archivists all benefit from this versatile tool. This underscores the significance of discovering valuable information within tables across a broad spectrum of diverse fields.

**Why Temporal Questions Answering?** Temporal questions require reasoning based on time-related information, falling into two main categories: explicit and implicit: (a.) Explicit temporal questions directly involve time-specific details like dates, years, or hours, demanding precise knowledge. For example, in Figure 1, question such as 'When was the Nedelcheva's was born?' is an explicit temporal question. (b.) Implicit temporal questions rely on temporal terms and relationships that indicate the temporal order or context of events or entities. These questions may include terms like "rank," "before," "after," "predecessor," "successor," and similar expressions. In such cases, the temporal dimension isn't explicitly stated (e.g., mention of year not in the table) but must be inferred (or extracted) from the given context with understanding of the relationships between elements. For instance, in Figure 1 'How many Bronze medals did Nedelcheva's won before 2013?' assumes an implicit understanding of the temporal sequence. In essence, addressing temporal questions involves comprehending and manipulating time-related information, whether explicit or implicit. This skill is vital in domains spanning historical research to natural language understanding, enabling effective reasoning about temporal aspects within data.

**Why a new Table QA dataset?** Current datasets such WIKITABLEQUESTIONS (Pasupat and Liang, 2015), SQUALL (Shi et al., 2020), FINQA (Chen et al., 2021d), TAT-QA (Zhu et al., 2021), HYBRIDQA (Chen et al., 2020c), FETAQA (Nan et al.,

2022), SequentialQA(SQA) (Iyyer et al., 2017), and WIKISQL (Zhong et al., 2017) for question-answering on table are limited in terms of both quantity and complexity of temporal questions. Table 31 show broad comparison across several tabular datasets. They fail to cover crucial aspects of temporal reasoning, necessitating the creation of a new manually curated dataset that specifically focuses on answering temporal questions related to tables. Our dataset, TEMPTABQA, can serve help train and evaluate models, aiding the development of more accurate and robust systems capable of effectively reasoning with temporal information in table-based contexts.

## 3 Our TEMPTABQA Dataset

We create a benchmark for answering temporal questions on entity-centric tables across domains.

### 3.1 Data Creation

**Table Domains Selection.** TEMPTABQA is built with Infobox tables from various Wikipedia articles across more than 90 categories. We focus on domains with time-related attributes in tables, particularly articles with Infoboxes containing temporal values. We analyze 1,208 Infobox templates[1] to compile a varied domain list, prioritizing entity tables with numerous temporal values like dates and times. Our analysis indicates that longer tables from popular and highly viewed articles contain a higher amount of temporal information, including an increased presence of temporal terms. As a result, these longer tables are deemed more appropriate for inclusion in TEMPTABQA. [2]

**Annotation Procedure.** To generate question-answer pairs from selected tables, we engage Amazon Mechanical Turk crowd-workers. MTurk annotators draft both the temporal question and answer based on a provided table. To ensure clarity, we instruct annotators to write clear, unambiguous, pronoun-free questions with grammatically complete answers. We also direct them to avoid yes/no questions and trivial questions that do not require temporal reasoning. This annotation approach ensures that the questions are challenging enough to evaluate models. We advise annotators to avoid

---

[1] https://en.wikipedia.org/wiki/Wikipedia: List_of_infoboxes
[2] We extract tables using BeautifulSoup4 and the Wikipedia extraction API.

repeating question patterns and instead use different starting tokens like "What," "When," "Whose," "How," "Where," "Who," "How many," etc., in order to prevent biased evaluation results. We instruct them to incorporate multiple unique rows from the table when formulating questions, including logical aspects that require reasoning to enhance complexity and variation. This evaluation approach ensures that the model's ability to reason across different parts of the table is assessed, and the questions are not overly basic. We encourage annotators to actively create unique questions, avoiding repetition and incorporating linguistic variation. This process fosters the generation of novel and interesting questions.

To answer the questions, we asked Turkers to provide brief responses in the form of phrases rather than full sentences. Additional information regarding Turking cost, annotator statistics, bonus and reward criteria, batch splitting, and other details are outlined in the appendix §D.

**Dealing with Annotation Bias.** Annotators may rely on personal biases, leading to position, selection, and popularity biases. They might also use repetitive question patterns and similar logical reasoning. To address these biases, we implemented measures such as: (1) *Diverse Table Categories*: Including tables from various categories in a single batch for diversity, with 12 distinct domains and no more than 3 tables from the same domain. (2) *Removal of Popular Rows*: Excluding frequent keys in entity tables, such as "Year Active," "Born," "Died," etc. (3) *Shuffling and Reordering*: Addressing position bias by shuffling table content and reordering subheadings like tournament titles, medal tallies, and awards. (4) *Mitigating Selection Bias*: Lessening selection bias by removing popular subsections, such as the "Olympics" section from an athlete table.

### 3.2 TEMPTABQA **Statistics and Analysis**

**Dataset.** Table 1 presents key metrics, including average row count, total unique tables, total questions, and average questions per table. We provide two test sets instead of one: the **Head** set with popular frequent domains, and the **Tail** set with rarer domains. Data split for train, development, head, and tail test sets are shown in Table 2.

**Questions.** Table 3 describes the composition and complexity of questions in our dataset. It presents the percentage of simple and complex

| Metric | Statistics | Metric | Statistics |
|---|---|---|---|
| #average rows | 22 | #unique tables | 1208 |
| #questions | 11454 | #question/table | 9.5 |

Table 1: TEMPTABQA dataset statistic.

| Dataset | #categories | #tables | #QA |
|---|---|---|---|
| train | 73 | 784 | 7680 |
| dev | 67 | 97 | 885 |
| head-set | 73 | 202 | 1851 |
| tail-set | 19 | 125 | 1038 |

Table 2: TEMPTABQA dataset splits statistic.

questions, taking into account multiple time-frame reasoning, the presence of single or multiple entities, and the inclusion of mathematical operations on temporal aspects. A question is deemed complex if it involves at least two simultaneous temporal reasoning steps from the categories of before-related, after-related, and in-between/during-related. Further details regarding these analyses, including mathematical operations such as min, max, count, average, difference, and comparison, can be found in Table 5.

| Question Type | Percent(%) | Question Type | Percent(%) |
|---|---|---|---|
| Simple | 57.81 | Complex | 42.19 |
| Multiple Entity | 47.90 | Single Entity | 52.10 |

Table 3: TEMPTABQA questions complexity.

We examine the required temporal intervals, including before, after, and present, as shown in Table 4. To categorize questions as *current*, we use keywords such as "until", "in", "during", "while", "at the same time", "meanwhile," "when", "since", and soon. *Past* questions contained keywords such as "before", "previous to", "prior to", "preceding", and soon., and *future* questions contained keywords such as "after", "following", "successor", "followed by", and soon.

| Type | Percent(%) | Interval | Percent(%) |
|---|---|---|---|
| #implicit | 63.23 | Past | 3.08 |
| #explicts | 36.76 | Future | 8.48 |
| #ordinal | 18.63 | Present | 66.64 |

Table 4: TEMPTABQA question reasoning type.

In addition, Table 4 also distinguishes between explicit and implicit temporal questions. Explicit questions mention a specific time or date, while implicit questions do not mention explicitly such temporal references. We also identified questions that used ordinal words or implied ranking or counting operations.

**Answers.** Table 6 breaks down answer types by counting examples whose answers are for several entity types: money, person, organization, location, percentage, and product.

| Operation | #QA | Operation | #QA | Operation | #QA |
|---|---|---|---|---|---|
| Maximum | 402 | Sum | 312 | Count | 3564 |
| Minimum | 377 | Average | 40 | | |
| Difference | 98 | Compare | 133 | | |

Table 5: TEMPTABQA questions mathematical operations.

| Analysis | Entity | #QA | Entity | #QA |
|---|---|---|---|---|
| | Money | 97 | Location | 411 |
| Entity Type | Person | 843 | Organization | 384 |
| | Percentage | 44 | Product | 57 |
| Analysis | Type | #QA | Type | #QA |
| | Count | 2130 | Ranking | 58 |
| Complexity | Boolean | 45 | Temporal | 4823 |
| | Age | 1085 | | |

Table 6: TEMPTABQA answer entity type and complexity.

Furthermore, we also evaluates answer complexity based on types such as count cardinal, ranking ordinal, boolean (Yes/No), temporal (Date/Time/Year), and age-related terms.

### 3.3 TEMPTABQA **Dataset Validation**

To ensure answer correctness in TEMPTABQA, we validate the development and two test sets by assigning three annotators to answer the questions based on the table. Annotators are instructed to provide concise yet comprehensive explanations to ensure accuracy and logical reasoning. The given instructions to annotators are as follows: (a.) *Use Table Information Only:* Annotators are instructed to rely solely on the table information, avoiding external knowledge except for common sense reasoning. (b.) *Clear, Concise, and Unambiguous Answers:* Annotators are asked to clear, concise, complete, and unambiguous answers and explanations, ensuring accuracy and clarity in the validation process. (c.) *Avoid Opinions or Assumptions:* To maintain objectivity and accuracy, annotators are instructed to refrain from including personal opinions or assumptions in their explanations. (d.) *Exclude Acronyms or Abbreviations:* To ensure clarity and avoid confusion, annotators are instructed to avoid using acronyms or abbreviations in their explanations. (e.) *Current Date and Year:* During the annotation process, we instructed annotators to consider December 2022 as the current month when answering questions that involve the present moment.

| Dataset | Majority Agreement | Human Accuracy |
|---|---|---|
| **Dev set** | 91.56 | 86.36 |
| **Head Test** | 93.62 | 86.17 |
| **Tail Test** | 89.73 | 86.47 |

Table 7: Data Validation Statistics, here both metric report the exact match.

TEMPTABQA **Filtering.** We use pre-processing to refine our training set, removing non-temporal and basic questions. The test and development sets undergo manual reviews by NLP experts after initial script-based filtering to maintain quality, focusing on complex temporal queries. We correct errors and prioritize questions that demand advanced reasoning, excluding those with direct answers or requiring external knowledge. Annotators were instructed to provide clear answers. However, some answers varied in format like "365 days" versus "one year". We made sure evaluation didn't penalize format differences, applying regex rules validated by human checks. For more details on filter refer to the appendix §D.

In the development set, less than 10% of questions, under 7% in the Head set, and under 11% in the Tail set were ambiguous, as shown in Table 7. Under 3% of questions were subjective. The most errors came from complex reasoning, whereas datetime errors were typically a year off. Around 82% of the annotated questions reach a clear majority consensus, demonstrating high agreement among annotators. For non-consensus questions, another review boosted agreement by 8-10%. By comparing the majority and gold answers, human accuracy was found to be 86%, as detailed in Table 7. See appendix §D, table 30 for fine-grained agreement.

## 4 Experimental Evaluation

We address the following research questions through our experiments: (a.) Is the new dataset TEMPTABQA challenging for existing models? (b.) Does finetuning on TEMPTABQA enhance model performance? (c.) Does providing few-shot examples and chain of thought reasoning benefit them? (d.) Is the performance on the tail domains worse than on the head domains?

**Evaluation:** We use the following metrics to evaluate the models: F1 score (F1), Exact Match (EM), Rouge 1 (R1) and 2 (R2), and Meteor (MET). For evaluation purposes, we treat December 2022 as the current month and year. To ensure models are aware of this, in all experiments, we add a new table row 'Current Date: December, 2022'.

**Models for Comparison.** Since most of the questions in TEMPTABQA require temporal and numerical reasoning to answer and are abstractive in nature, we mostly use decoder-only models (except for BART which are encoder-decoder models). We consider the following models: (a.) **Fine-tuned model**: BART-Large, T5-XL, and Flan-T5-XL, along with smaller versions, all fine-tuned on TEMPTABQA. (b.) **Zero-shot LLM**: T5-XXL, Flan-T5-XXL, LLaMA-2, GPT-3.5 and 4, and PaLM along with smaller versions, without fine-tuning. (c.) **Few-shot LLM**: Same models as with zero-shot but in few shot settings with three reference examples. (d.) **Few-shot LLM with Chain of Thoughts:** Similar to the few-shot setup, but with chain-of-thought (Wei et al., 2022) reasoning included with examples. [3]

For additional details on the these models, including hyperparameter information, please refer to the appendix §C.

## 4.1 Our Findings: Results and Analysis

Table 8, 9, 10, 11 show the zero-shot, fine tuned, few-shot w/o chain of thoughts and few-shot with chain of thoughts prompting models performance.

**TEMPTABQA is Challenging.** The dataset presents an challenging task, with all models performing significantly worse than the human, refer to Table 8, 9, 10, 11. Even our top-performing model, GPT-4 with Chain of Thought prompting, lags behind humans by 13.19 and 20.61 F1 points on the Head and Tail sets, respectively. Additionally, our best fine-tuned model, Flan-T5-XL, trails even further behind, with a margin of 31.75 and 32.58 F1 points on the Head and Tail sets. [4] Furthermore, the GPT model consistently outperforms other models, such as Flan-T5 and T5, in both zero-shot and few-shot settings. Turning tables into knowledge graphs (+KG) [5] results in the model's superior performance compared to conventional linearization methods.

**Fine-tuning Helps.** Our findings, in Table 9, highlight the significant advantages of fine-tuning medium-scale models. Remarkably, fine-tuned Flan-T5-XL models outperform the non-fine-tuned

[3] We didn't include TabT5 due to proprietary industry restrictions (not published in the open source dataset).
[4] Due to resource limitations we didn't fine-tuning models larger than XL size.
[5] We use GPT-4 with human in the loop to convert table to Knowledge Graph.

| Model | Size | F1 | EM | R1 | R2 | MET |
|---|---|---|---|---|---|---|
| **Head Domain** | | | | | | |
| **T5** | L | 35.51 | 33.93 | 35.73 | 35.67 | 23.97 |
| | XL | 35.51 | 33.93 | 35.73 | 35.67 | 27.07 |
| | XXL | 38.08 | 36.77 | 38.08 | 38.05 | 25.86 |
| **Flan-T5** | L | 33.81 | 32.04 | 33.91 | 33.87 | 22.43 |
| | XL | 41.80 | 40.72 | 41.83 | 41.8 | 27.17 |
| | XXL | 43.29 | 41.87 | 43.41 | 43.40 | 27.78 |
| **LLaMA** | 2 | 47.90 | 40.73 | 48.36 | 48.28 | 33.62 |
| **GPT** | 3.5 | 53.38 | 49.03 | 53.64 | 53.56 | 39.09 |
| | 4 | 69.97 | 65.17 | 70.24 | 70.22 | 50.33 |
| **+KG** | 4 | **72.24** | **68.02** | **72.98** | **72.86** | **52.10** |
| **PaLM** | 2 | 69.05 | 66.82 | 69.00 | 68.91 | 42.32 |
| **Human** | | 87.49 | 86.17 | 87.61 | 87.61 | 58.87 |
| **Tail Domain** | | | | | | |
| **T5** | L | 28.02 | 25.41 | 29.01 | 28.96 | 18.45 |
| | XL | 31.39 | 28.72 | 32.39 | 32.32 | 20.45 |
| | XXL | 30.12 | 27.65 | 30.92 | 30.88 | 19.82 |
| **Flan-T5** | L | 29.06 | 26.29 | 29.98 | 29.92 | 18.25 |
| | XL | 36.54 | 34.76 | 37.65 | 37.58 | 22.35 |
| | XXL | 38.68 | 36.81 | 39.92 | 39.88 | 23.53 |
| **LLaMA** | 2 | 39.75 | 31.59 | 40.68 | 40.65 | 327.02 |
| **GPT** | 3.5 | 47.81 | 43.04 | 49.22 | 49.13 | 33.81 |
| | 4 | 60.54 | 55.21 | 62.17 | 62.15 | 42.49 |
| **+KG** | 4 | **62.80** | **57.80** | **64.18** | **64.16** | **43.99** |
| **PaLM** | 2 | 61.64 | 58.38 | 63.14 | 63.07 | 37.23 |
| **Human** | | 87.82 | 86.47 | 87.97 | 87.94 | 57.26 |

Table 8: Zero Shot Setting.

Flan-T5-XXL model, which is even larger in size, in various few-shot scenarios, including chain-of-thought prompting, by impressive margins of 13.79 and 17.18 F1 points. However, when compared to the GPT models, particularly in few-shot scenarios with chain-of-thought prompting, the fine-tuned models fall short by 18.56 and 11.97 on the Head and Tail sets respectively.

| Model | Size | F1 | EM | R1 | R2 | MET |
|---|---|---|---|---|---|---|
| **Head Domain** | | | | | | |
| **BART** | B | 38.06 | 26.72 | 38.69 | 38.68 | 26.58 |
| | L | 45.68 | 34.56 | 46.35 | 46.32 | 29.08 |
| **T5** | B | 42.37 | 35.40 | 42.90 | 42.84 | 28.26 |
| | L | 49.48 | 39.03 | 50.19 | 50.12 | 34.07 |
| | XL | 52.82 | 42.16 | 53.49 | 53.42 | 36.42 |
| **Flan-T5** | B | 43.24 | 37.03 | 43.85 | 43.77 | 28.39 |
| | L | 47.86 | 39.35 | 48.49 | 48.41 | 29.36 |
| | XL | **55.74** | **45.56** | **56.46** | **56.41** | **38.74** |
| **Human** | | 87.49 | 86.17 | 87.61 | 87.61 | 58.87 |
| **Tail Domain** | | | | | | |
| **BART** | B | 35.62 | 24.44 | 36.74 | 36.68 | 24.31 |
| | L | 41.99 | 30.77 | 43.16 | 43.08 | 26.01 |
| **T5** | B | 36.76 | 29.89 | 37.55 | 37.52 | 22.97 |
| | L | 44.45 | 35.15 | 45.95 | 45.75 | 28.89 |
| | XL | 51.61 | 41.19 | 53.42 | 53.35 | 34.61 |
| **Flan-T5** | B | 38.20 | 31.84 | 39.34 | 39.28 | 23.86 |
| | L | 41.83 | 32.91 | 43.19 | 42.99 | 23.23 |
| | XL | **55.24** | **45.08** | **56.94** | **56.91** | **37.11** |
| **Human** | | 87.82 | 86.47 | 87.97 | 87.94 | 57.26 |

Table 9: Instruction Fine-tune Models.

**Few-shot (w CoT) > few-shot (w/o CoT) > zero-shot).** Tables 10 and 11 shows that few-shot models outperform their zero-shot counterparts. For instance, GPT-4 shows a gain of 2.0 and 2.23 F1 points on the Head and Tail sets, respectively, in the few-shot version compared to the zero-shot version. This trend is consistent across models like Flan-T5 and T5, regardless of model size. Notably, larger model sizes (L to XL to XXL) yield improved performance. Furthermore, incorporating chain-of-thought prompting provides an additional boost to the model's performance. Furthermore, linearization outperforms knowledge graphs.

| Model | Size | F1 | EM | R1 | R2 | MET |
|---|---|---|---|---|---|---|
| | | **Head Domain** | | | | |
| | L | 35.79 | 34.14 | 35.89 | 35.83 | 23.95 |
| Flan-T5 | XL | 41.64 | 40.50 | 41.70 | 41.67 | 27.07 |
| | XXL | 41.06 | 39.35 | 41.23 | 41.21 | 26.96 |
| LLaMA | 2 | 53.57 | 53.57 | 53.67 | 53.58 | 37.95 |
| GPT | 3.5 | 57.35 | 53.34 | 57.65 | 57.54 | 42.55 |
| | 4 | **71.97** | **67.07** | **72.15** | **72.10** | **51.60** |
| +KG | 4 | 70.48 | 65.69 | 70.71 | 70.66 | 50.32 |
| PaLM | 2 | 68.84 | 66.40 | 68.96 | 68.90 | 42.78 |
| Human | | 87.49 | 86.17 | 87.61 | 87.61 | 58.87 |
| | | **Tail Domain** | | | | |
| | L | 29.70 | 26.78 | 30.52 | 30.45 | 18.88 |
| Flan-T5 | XL | 36.48 | 34.86 | 37.82 | 37.75 | 22.38 |
| | XXL | 36.48 | 34.86 | 37.82 | 37.75 | 22.51 |
| LLaMA | 2 | 46.01 | 37.66 | 46.70 | 46.67 | 31.76 |
| GPT | 3.5 | 53.43 | 49.37 | 54.24 | 54.12 | 39.06 |
| | 4 | **62.77** | **57.94** | **64.37** | **64.34** | **44.21** |
| +KG | 4 | 61.99 | 57.42 | 63.55 | 63.52 | 43.67 |
| PaLM | 2 | 59.94 | 57.42 | 61.46 | 61.39 | 36.65 |
| Human | | 87.82 | 86.47 | 87.97 | 87.94 | 57.26 |

Table 10: Few Shot w/o Chain of Thought Prompting.

| Model | Size | F1 | EM | R1 | R2 | MET |
|---|---|---|---|---|---|---|
| | | **Head Domain** | | | | |
| | L | 35.20 | 32.88 | 35.21 | 35.16 | 25.32 |
| Flan-T5 | XL | 38.31 | 35.46 | 38.52 | 38.51 | 25.84 |
| | XXL | 41.95 | 39.61 | 42.02 | 41.94 | 30.27 |
| LLaMA | 2 | 50.21 | 44.02 | 50.45 | 50.44 | 35.88 |
| GPT | 3.5 | 62.15 | 56.13 | 62.63 | 62.58 | 44.63 |
| | 4 | **74.30** | **68.96** | **74.49** | **74.47** | **53.07** |
| +KG | 4 | 72.82 | 67.37 | 73.11 | 73.09 | 51.95 |
| PaLM | 2 | 68.41 | 64.07 | 68.55 | 68.47 | 44.38 |
| Human | | 87.49 | 86.17 | 87.61 | 87.61 | 58.87 |
| | | **Tail Domain** | | | | |
| | L | 31.22 | 28.43 | 31.47 | 31.34 | 22.13 |
| Flan-T5 | XL | 33.12 | 29.79 | 34.17 | 34.14 | 21.50 |
| | XXL | 38.06 | 34.86 | 38.81 | 38.67 | 27.07 |
| LLaMA | 2 | 46.07 | 39.26 | 46.77 | 46.72 | 31.63 |
| GPT | 3.5 | 55.84 | 50.05 | 57.32 | 57.25 | 39.60 |
| | 4 | **67.21** | **61.54** | **68.66** | **68.64** | **47.50** |
| +KG | 4 | 64.67 | 58.95 | 66.22 | 66.19 | 45.45 |
| PaLM | 2 | 61.56 | 55.87 | 62.94 | 62.81 | 39.32 |
| Human | | 87.82 | 86.47 | 87.97 | 87.94 | 57.26 |

Table 11: Few Shot with Chain of Thought Prompting.

**Head vs. Tail domain.** Our observations reveal that the tail set posed greater challenges for all models across various settings, while humans achieved similar performance on both sets. Models face greater challenges with tail tables in contrast to head tables. For instance, even the top-performing model, GPT-4, showed a difference of around 9.20 F1 points, performing better on the Head set in zero-shot scenarios. However, this performance gap diminished with few-shot learning and chain-of-thought reasoning. In few-shot scenarios with chain-of-thought prompting, the gap reduced to 7.09 F1 points This phenomenon mainly results from knowledge transfer between less common and widely recognized sports tables. The head tables exhibit many common attributes and pose similar types of questions, in contrast to the rare tables

## 5 Analysis Breakdown of Performance

In our analysis, we examine the results (exact match) of our best model, GPT-4 few-shot with chain of thought, alongside human performance.

**Question Types.** We categorize questions based on their types: starting with "what," "where," "when," "how," or "quantity" (also known as "how many"). The evaluation of the GPT-4 model's performance (exact match) compared to humans is presented in Table 12.

| Question Type | Head Set | | | Tail Set | | |
|---|---|---|---|---|---|---|
| | # | Human | GPT-4 | # | Human | GPT-4 |
| what | 470 | 87.23 | 70.64 | 326 | 86.20 | 64.11 |
| where | 22 | 95.45 | 90.91 | 12 | 83.33 | 50.00 |
| who | 80 | 88.75 | 63.75 | 44 | 70.45 | 34.09 |
| when | 264 | 90.87 | 74.24 | 102 | 91.18 | 69.31 |
| how many | 588 | 80.27 | 70.75 | 378 | 81.48 | 67.46 |
| how much | 151 | 81.58 | 62.91 | 105 | 81.90 | 65.71 |

Table 12: Performance comparison Question Types

*Analysis.* Humans consistently outperform the model in all scenarios, with a notable performance disparity in the tail domain. The model demonstrates relatively stronger performance in answering "Where" and "How Much" questions compared to other types. However, it faces challenges in tackling "What," "Who," and "When" questions, resulting in lower performance. We observe that humans handle "Where" questions with the least difficulty and struggle the most with "How Many" questions. Conversely, the model encounters significant challenges with "Who" questions and performs relatively better with "Where" question types.

**Reasoning Operation.** To answer the questions, various analytical reasoning operations are involved, such as maximum, minimum, counting, summation, average, difference, and comparison. Table 13 provides a evaluation of the GPT-4 model's performance (exact match) compared to human performance, focusing on these operations.

| Reasoning Operation | Head Set | | | Tail Set | | |
|---|---|---|---|---|---|---|
| | # | Human | GPT-4 | # | Human | GPT-4 |
| Maximum | 89 | 95.51 | 79.78 | 42 | 83.34 | 69.05 |
| Minimum | 102 | 87.25 | 67.65 | 38 | 86.84 | 71.05 |
| Counting | 603 | 80.41 | 73.47 | 375 | 82.93 | 70.41 |
| Summation | 44 | 70.45 | 52.27 | 38 | 68.42 | 42.11 |
| Difference | 16 | 62.51 | 43.75 | 11 | 72.72 | 54.54 |
| Comparison | 21 | 80.95 | 66.67 | 13 | 69.23 | 53.85 |

Table 13: Performance comparison w.r.t. Operations.

*Analysis.* Once again, it is evident that humans consistently outperform the model in all types of operations, particularly in challenging tasks. Furthermore, our observations reveal that the model demonstrates relatively stronger performance in analytical reasoning tasks like "maximum" and "counting" compared to other types of tasks. However, it faces significant challenges in tasks such as "minimum," "difference," and "comparison," resulting in lower performance levels. Overall, both humans and the model excel in "maximum" tasks while struggling with "difference" and "summation" tasks. Additionally, the model's performance in "minimum" and "comparison" tasks falls short compared to human performance, indicating its limitations in these areas.

**Explicit or Implicit.** Our analysis compares the performance of humans and the best model in answering explicit and implicit time-related questions. Explicit questions directly mention time and can be found in the table, while implicit questions require inferring the time from the table information. Table 15 showcases the model's performance on both question types.

| Answer Type | Head Set | | | Tail Set | | |
|---|---|---|---|---|---|---|
| | # | Human | GPT-4 | # | Human | GPT-4 |
| explicit | 565 | 85.31 | 68.5 | 296 | 81.65 | 57.43 |
| implicit | 1018 | 84.38 | 71.71 | 686 | 87.87 | 67.64 |

Table 14: Performance comparison Answer Types.

*Analysis.* The model demonstrates better performance in implicit temporal reasoning compared to explicit temporal reasoning. As earlier model struggles more with rare and infrequent questions in the tail domain. Implicit temporal reasoning questions are more prevalent, with a greater performance difference between the two types observed in the tail set. Notably, humans also struggle more with explicit questions compared to implicit ones, likely due to increased complexity and advanced mathematical reasoning requirements. Explicit questions demand deeper understanding and precise reasoning, explicitly stating specific temporal information, while implicit questions rely more on contextual reasoning and inference, allowing the model to leverage broader table information.

**Answer Types.** We analyze the entity or common noun type of the answer. Answer categories include age (gap or sum), count, monetary terms, ordinal numbers, organization names, percentages, person names, place names, product specifics, temporal entities (date, time, day), Boolean (yes/no, true/false, comparison), or unknown (not any specific type). Table 15 presents the model's performance based on the type of answer entity.

| Entity Type | Head Set | | | Tail Set | | |
|---|---|---|---|---|---|---|
| | # | Human | GPT-4 | # | Human | GPT-4 |
| Boolean | 2 | 100 | 50.00 | 7 | 57.14 | 42.86 |
| Temporal | 736 | 83.7 | 69.93 | 411 | 81.11 | 64.23 |
| Count | 341 | 83.87 | 75.37 | 245 | 86.94 | 75.51 |
| Age | 133 | 83.46 | 56.72 | 85 | 91.67 | 60 |
| Money | 17 | 82.35 | 64.71 | 3 | 66.67 | 66.67 |
| Percentage | 8 | 62.5 | 37.5 | 6 | 16.67 | 33.33 |
| Ordinal | 6 | 66.67 | 50 | 1 | 0 | 0 |
| Place | 47 | 97.87 | 87.23 | 24 | 87.5 | 62.5 |
| Person | 76 | 89.47 | 65.79 | 43 | 69.77 | 32.56 |
| Organization | 69 | 89.86 | 82.61 | 43 | 95.35 | 60.47 |
| Unknown | 146 | 85.62 | 69.86 | 107 | 83.18 | 60.75 |

Table 15: Performance comparison Entity Types.

*Analysis.* Our analysis reveals that the model struggles with calculating age gaps, boolean, place, and person-related questions, in contrast to count-related questions. Similar to previous findings, both the model and humans perform better on frequent head domain tables compared to tail domain tables. However, regardless of table type, both humans and the model encounter difficulties with percentages and ordinals. The model's performance is notably weaker in age gap calculations, boolean, place, and person-related questions, while exhibiting better performance in count-related questions. Additionally, both humans and the model face challenges with percentages and ordinals across table domains. For GPT-3.5 analysis, refer to appendix §A. Category-specific analysis based on table domain is in appendix §B.

## 6 Comparison with Related Work

**Tabular Datasets and Models.** Recent studies have explored various NLP tasks on semi-structured tabular data, including tabular natural language inference, fact verification (Chen et al., 2020b; Gupta et al., 2020; Zhang and Balog, 2019), question answering, semantic parsing (Zhang and Balog, 2020; Zhang et al., 2020b; Pasupat and Liang, 2015; Krishnamurthy et al., 2017; Abbas et al., 2016; Sun et al., 2016; Chen et al., 2020c; Lin et al., 2020; Zayats et al., 2021; Oguz et al., 2020; Chen et al., 2021b; Iyyer et al., 2017), and table-to-text generation (Parikh et al., 2020; Li et al., 2021; Nan et al., 2021; Yoran et al., 2021; Chen et al., 2020a).

Various strategies have been proposed to represent Wikipedia relational tables, including Table2vec (Deng et al., 2019), TAPAS (Herzig et al., 2020), TaBERT (Yin et al., 2020), TabStruc (Zhang et al., 2020a), TABBIE (Iida et al., 2021), TabGCN (Pramanick and Bhattacharya, 2021), and RCI (Glass et al., 2021). Pre-training methods have also been studied to improve tabular inference (Yu et al., 2018, 2021; Eisenschlos et al., 2020; Neeraja et al., 2021). Recent shared tasks like SemEval'21 Task 9 (Wang et al., 2021) and FEVEROUS'21 shared task (Aly et al., 2021) have further explored these areas.

In comparision to prior work, TEMPTABQA centers on temporal question answering within entity-centric tables, an untapped domain. While most datasets lean towards non-temporal queries, they seldom address temporal aspects and lack a grounding in the common sense and the necessary world knowledge. These datasets predominantly emphasize arithmetic reasoning using SQL in structured formats, overlooking the nuanced semi-structured Infobox-style tables rich in common sense.

**Temporal Datasets and Models.** Several temporal question answering datasets have been introduced recently. These include TIME-SENSITIVE-QA (Chen et al., 2021c) and TORQUE (Ning et al., 2020), which are entity-specific reading comprehension datasets with time-sensitive questions derived from Wikipedia paragraphs. TEMPQA-WD (Neelam et al., 2022), CRONQUESTIONS (Saxena et al., 2021), and TEMPQUESTIONS (Jia et al., 2018a) are question answering datasets focusing on knowledge graph embeddings with temporal links. Additionally, there are open-domain (Zhang and

Choi, 2021) and cloze-form (Dhingra et al., 2022) question answering tasks, as well as event-centric datasets (Ning et al., 2018; Wen et al., 2021; Chen et al., 2021a) that explore temporal QA.

In terms of modeling, there are temporally tuned language models trained on knowledge-based question answering datasets such as CRONKBQA (Saxena et al., 2021), TEQUILA (Jia et al., 2018b), EXAQT (Jia et al., 2021a), OTR-QA (Shang et al., 2021), and TEMPOQR (Mavromatis et al., 2021), among others. (Kannen et al., 2022) suggest a targeted approach to extract temporal facts when traditional KBQA methods fail to retrieve them from the knowledge base. Some methods incorporate temporal aspects during masked language model pre-training (Dhingra et al., 2022; Iv et al., 2022), rather than fine-tuning on downstream NLI tasks. In comparison to prior work, TEMPTABQA focuses on temporal question answering specifically on entity-centric tables, while most existing studies address non-tabular datasets.

## 7 Conclusion

In conclusion, this study addresses the effectiveness of current NLP systems in reasoning about temporal information in semi-structured data, specifically Infobox tables. We introduce the task of temporal question answering on semi-structured tables and present the TEMPTABQA dataset, consisting of 11,454 question-answer pairs from 1,208 Wikipedia Infobox tables across varied domains. Evaluating state-of-the-art models on this dataset reveals significant gaps compared to human performance, exceeding 13.5 F1 points. These findings emphasize the need for advancements in temporal reasoning capabilities of NLP models. The TEMPTABQA dataset serves as a challenging benchmark to enhance temporal reasoning in NLP models.

**Future Directions.** From our analysis, we suggest future avenues in temporal query answering: (a) **Diverse Structures:** Expand temporal queries to various table structures, like hybrid compositions (text, table, image). (b) **Dynamic Queries:** Examine evolving tables across a consistent timeline. (c) **Open Domain Queries:** Merge retrieval, extraction, understanding, and temporal reasoning into one framework. (d) **Reasoning with LLMs:** Tailor large language models (LLMs) for table-specific temporal logic, with more on this in Appendix H. Advanced prompts remain a potential area of exploration.

## Limitations

First, it focuses solely on entity-centric tables from Wikipedia, excluding non-Infobox tables and other relevant sources. Exploring a broader range of table types would be valuable. Second, despite our efforts to ensure unbiased table selection and dataset annotation, inadvertent bias leakage is possible, potentially affecting results.

Third, due to limited computational capacity, we could only fine-tune models using large sizes (XL), not extra-large (XXL). One idea that could be explore here is using Parameter Efficient Fine tuning (PEFT) (Mangrulkar et al., 2022). Incorporating more open-source Language Models (LLMs) would enhance our understanding of temporal reasoning capabilities. Lastly, our work primarily targets the English language, while exploring multilingual settings would increase applicability and generalizability. These limitations present opportunities for future research and expansion in this field.

## Aknowledgement

The authors express their gratitude to Bloomberg's AI Engineering team, particularly Edgar Meij and Prabhanjan Kambadur, for their invaluable feedback and guidance. We are also thankful for the valuable insights provided by Ellen Riloff, Dan Roth, and the Utah NLP group. Special thanks to Dibyakanti Kumar and Manasvi Kundalia for their contributions. Vivek Gupta acknowledges the support received from Bloomberg's Data Science Ph.D. Fellowship and the Cognitive Computation Group at the University of Pennsylvania. This work is partially supported by NSF grants #1801446, #1822877, #2007398 and #2129111. The views and conclusions contained herein are those of the authors and should not be interpreted as necessarily representing the official policies of any government agency. Lastly, we extend our appreciation to the reviewing team for their insightful comments.

## Ethics Statement

The dataset in this study is designed for research on temporal question answering with entity-centric tables. It should be strictly used for research purposes, not for other applications. We have diligently created the dataset to minimize bias during table selection and question-answer generation. However, inadvertent bias leakage is still possible, so thorough examination is crucial for uses beyond the intended research scope.

To ensure fairness, we provided fair wages to MTurk crowd workers and conducted three pilot studies to estimate task completion time accurately. We also plan to release a datasheet, full annotation template, and other resources for data and model openness. Emphasizing openness enables issue identification and correction, allowing continuous improvement based on community feedback. These measures promote transparency and facilitate further advancements in the field.

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

## A  Analysis Breakdown: GPT-3.5 model

**GPT-3.5 Model:**  Table 16, 17, 18, 19, 20, 21 represent analysis of GPT-3.5 on various aspects such as question type, reasoning operation, explicit and implicit , entity type and category wise (tail and head), respectively.

## B  Analysis Breakdown: GPT-4 vs Human

**GPT-4 vs. Human:**  Table 22, 23, 24, 25 represent analysis of GPT-4 and Human on various aspects such as question type, reasoning operation, explicit and implicit , entity type.

We also consider what the performance of model as comapred to Human on examples on particular table/article domain. We consider coarse grained categories for comparison. Table 26, 27 and 28, 29 shows the model head and tail set performance based on table domains for coarse and fine-gained setting.

| Type | # | F1 | EM | R1 | R2 | MET |
|------|---|----|----|----|----|-----|
| | | **Head** | | | | |
| what | 470 | 59.34 | 57.87 | 59.43 | 59.37 | 40.92 |
| where | 22 | 88.41 | 86.36 | 89.81 | 89.81 | 65.5 |
| who | 80 | 61.59 | 58.75 | 61.68 | 61.68 | 55.28 |
| when | 263 | 69.51 | 68.06 | 69.42 | 69.39 | 49.79 |
| quantity | 588 | 58.88 | 56.97 | 58.91 | 58.91 | 45.22 |
| how | 152 | 50.02 | 48.03 | 50.36 | 50.36 | 46.65 |
| | | **Tail** | | | | |
| what | 326 | 55.67 | 54.29 | 56.54 | 56.42 | 38.18 |
| where | 12 | 55.26 | 50.0 | 55.25 | 55.25 | 47.56 |
| who | 44 | 36.31 | 31.82 | 36.39 | 36.39 | 35.96 |
| when | 101 | 66.12 | 64.36 | 66.11 | 66.11 | 50.25 |
| quantity | 378 | 57.29 | 55.29 | 57.4 | 57.33 | 39.76 |
| how | 105 | 56.87 | 55.24 | 56.99 | 56.99 | 47.83 |

Table 16: Performance w.r.t Question Type with GPT-3.5 few shots (with chain of thought prompting).

| Op. | F1 | EM | R1 | R2 | MET |
|-----|----|----|----|----|-----|
| | **Head** | | | | |
| max | 68.74 | 67.42 | 68.69 | 68.69 | 48.89 |
| min | 55.78 | 53.92 | 56.19 | 56.15 | 47.53 |
| count | 61.11 | 59.3 | 61.1 | 61.1 | 45.64 |
| sum | 36.02 | 34.09 | 36.22 | 36.22 | 25.59 |
| avg. | 100.0 | 100.0 | 100.0 | 100.0 | 93.75 |
| dif. | 45.44 | 43.75 | 45.62 | 45.62 | 41.15 |
| com. | 48.48 | 47.62 | 48.64 | 48.64 | 35.19 |
| | **Tail** | | | | |
| max | 66.18 | 64.29 | 66.81 | 66.81 | 47.93 |
| min | 52.32 | 50.0 | 52.52 | 52.52 | 38.45 |
| count | 59.74 | 57.87 | 59.79 | 59.72 | 40.61 |
| sum | 43.48 | 39.47 | 43.71 | 43.52 | 32.36 |
| avg. | 34.33 | 25.0 | 34.58 | 33.33 | 39.66 |
| dif. | 56.8 | 54.55 | 57.4 | 57.4 | 51.08 |
| com. | 54.99 | 53.85 | 54.83 | 54.83 | 35.46 |

Table 17: Performance w.r.t Reasoning Operation with GPT-3.5 few shots (with chain of thought prompting).

| Domain | # | EM | Domain | # | EM |
|--------|---|----|--------|---|----|
| person | 66 | 59.09 | sports | 717 | 56.49 |
| cricket team | 18 | 61.11 | history | 27 | 33.33 |
| aircraft | 24 | 66.67 | fighter | 9 | 33.33 |
| finance | 31 | 41.94 | court | 21 | 71.43 |
| musician | 42 | 59.52 | art | 30 | 83.33 |
| nobel | 39 | 56.41 | country | 9 | 44.44 |
| space | 51 | 49.02 | railway | 15 | 80.0 |
| company | 21 | 52.38 | website | 3 | 33.33 |
| university | 18 | 72.22 | monument | 27 | 77.78 |
| event | 5 | 40.0 | book | 15 | 86.67 |
| church | 15 | 60.0 | leaders | 27 | 66.67 |
| office holders | 27 | 48.15 | music | 22 | 77.27 |
| war | 34 | 38.24 | conflicts | 14 | 14.29 |
| concert | 27 | 62.96 | disaster | 18 | 55.56 |
| song | 18 | 77.78 | movie | 24 | 66.67 |
| rail line | 5 | 60.0 | character | 27 | 70.37 |
| ships | 15 | 53.33 | agency | 16 | 75.0 |
| board game | 48 | 75.0 | NFT | 18 | 66.67 |
| NCT | 18 | 72.22 | | | |

Table 18: Head Set coarse-gained category-Wise Results with GPT-3.5 few-shot (with Chain of Thought).

| Type | F1 | EM | R1 | R2 | MET |
|---|---|---|---|---|---|
| | | | **Head** | | |
| exp. | 59.44 | 56.99 | 0.6 | 59.57 | 48.24 |
| imp. | 61.08 | 59.72 | 0.61 | 61.1 | 44.16 |
| | | | **Tail** | | |
| exp. | 52.50 | 50.0 | 0.53 | 52.81 | 38.64 |
| imp. | 58.51 | 56.85 | 0.59 | 58.76 | 42.05 |

Table 19: Performance w.r.t Implicit/Explicit Type with GPT-3.5 few shots (with chain of thought prompting).

| Type | # | F1 | EM | R1 | R2 | MET |
|---|---|---|---|---|---|---|
| | | | | **Head** | | |
| Temporal | 736 | 65.44 | 63.32 | 65.49 | 65.47 | 55.4 |
| Count | 341 | 55.91 | 55.13 | 55.93 | 55.93 | 30.0 |
| Age | 133 | 45.13 | 43.61 | 45.09 | 45.05 | 24.25 |
| Money | 17 | 55.38 | 52.94 | 56.25 | 56.25 | 54.36 |
| Percentage | 8 | 6.13 | 0.0 | 5.36 | 5.36 | 6.95 |
| Ordinal | 6 | 33.33 | 33.33 | 33.33 | 33.33 | 20.14 |
| Place | 47 | 69.68 | 68.09 | 70.28 | 70.28 | 45.74 |
| Person | 76 | 64.37 | 61.84 | 64.52 | 64.52 | 57.88 |
| Organization | 69 | 62.13 | 60.87 | 62.28 | 62.08 | 52.49 |
| Product | 2 | 100.0 | 100.0 | 100.0 | 100.0 | 71.88 |
| Unknown | 146 | 58.78 | 56.85 | 59.05 | 59.0 | 44.66 |
| Boolean | 2 | 50.0 | 50.0 | 50.0 | 50.0 | 25.0 |
| | | | | **Tail** | | |
| Temporal | 412 | 58.38 | 55.34 | 58.56 | 58.47 | 52.04 |
| Count | 245 | 58.35 | 57.96 | 58.45 | 58.37 | 30.32 |
| Age | 84 | 61.37 | 60.71 | 61.54 | 61.47 | 31.92 |
| Money | 3 | 69.84 | 66.67 | 72.46 | 72.46 | 65.83 |
| Percentage | 6 | 35.0 | 33.33 | 37.19 | 37.19 | 20.61 |
| Ordinal | 1 | 0.0 | 0.0 | 0.0 | 0.0 | 0.0 |
| Place | 24 | 64.04 | 62.5 | 63.89 | 63.89 | 41.34 |
| Person | 43 | 34.82 | 30.23 | 34.91 | 34.91 | 34.62 |
| Organization | 43 | 44.85 | 44.19 | 47.85 | 47.85 | 28.17 |
| Product | 7 | 85.71 | 85.71 | 85.71 | 85.71 | 42.86 |
| Unknown | 107 | 54.49 | 53.27 | 55.24 | 55.17 | 39.14 |
| Boolean | 7 | 51.03 | 42.86 | 51.37 | 50.27 | 32.66 |

Table 20: Performance w.r.t Entity Type with GPT-3.5 few shots (with chain of thought prompting).

| Domain | # | EM | Domain | # | EM |
|---|---|---|---|---|---|
| sports | 500 | 61.2 | party | 72 | 52.78 |
| time zone | 15 | 53.33 | holiday | 96 | 40.62 |
| ships | 75 | 57.33 | aircraft | 6 | 50.0 |
| organization | 6 | 66.67 | disaster | 36 | 41.67 |
| war | 63 | 49.21 | army | 75 | 42.67 |
| planet | 6 | 50.0 | diseases | 32 | 50.0 |

Table 21: Tail set coarse-gained category-Wise Results with GPT-3.5 few-shot (with Chain of Thought).

| Type | Human F1 | GPT-4 F1 | Human R1 | GPT-4 R1 | Human R2 | GPT-4 R2 | Human MET | GPT-4 MET |
|---|---|---|---|---|---|---|---|---|
| | | | | **Tail Domain** | | | | |
| | | | | **Head Set** | | | | |
| what | 88.48 | 72.46 | 88.43 | 72.6 | 88.3 | 72.5 | 59.13 | 50.57 |
| where | 95.95 | 92.12 | 95.86 | 93.42 | 95.86 | 93.42 | 67.45 | 69.17 |
| who | 91.74 | 67.47 | 91.61 | 67.31 | 91.61 | 67.31 | 78.64 | 60.99 |
| when | 91.56 | 76.03 | 91.56 | 75.97 | 91.37 | 75.97 | 59.84 | 53.06 |
| quantity | 82.24 | 72.71 | 81.6 | 72.7 | 81.6 | 72.66 | 49.05 | 53.58 |
| how | 86.93 | 64.84 | 87.45 | 65.06 | 87.45 | 65.01 | 74.93 | 59.31 |
| | | | | **Tail Set** | | | | |
| what | 88.17 | 65.91 | 88.21 | 67.02 | 88.14 | 66.82 | 55.98 | 45.32 |
| where | 83.33 | 55.28 | 83.33 | 55.45 | 83.33 | 55.45 | 56.62 | 47.0 |
| who | 75.66 | 40.08 | 75.66 | 40.45 | 75.66 | 40.23 | 67.12 | 40.22 |
| when | 93.12 | 71.28 | 93.12 | 71.25 | 93.12 | 71.25 | 65.38 | 55.12 |
| quantity | 84.39 | 69.43 | 84.51 | 69.55 | 84.51 | 69.51 | 48.11 | 47.19 |
| how | 85.41 | 67.46 | 85.41 | 67.78 | 85.41 | 67.78 | 73.16 | 59.07 |

Table 22: Comparison between Human and GPT-4 w.r.t. question type

| Op. | Human F1 | GPT-4 F1 | Human R1 | GPT-4 R1 | Human R2 | GPT-4 R2 | Human MET | GPT-4 MET |
|---|---|---|---|---|---|---|---|---|
| | | | | **Head Set** | | | | |
| max | 95.51 | 80.92 | 95.51 | 80.91 | 95.51 | 80.84 | 61.62 | 57.11 |
| min | 88.24 | 70.74 | 88.24 | 70.87 | 88.24 | 70.66 | 68.4 | 59.58 |
| count | 82.25 | 75.27 | 81.63 | 75.22 | 81.63 | 75.19 | 47.29 | 54.05 |
| sum | 78.08 | 54.33 | 75.81 | 54.32 | 75.81 | 54.18 | 43.1 | 36.88 |
| avg. | 100.0 | 100.0 | 100.0 | 100.0 | 100.0 | 100.0 | 93.75 | 93.75 |
| dif. | 71.25 | 46.63 | 71.25 | 46.41 | 71.25 | 46.41 | 54.87 | 39.42 |
| com. | 80.95 | 70.18 | 82.14 | 71.24 | 82.14 | 71.24 | 54.37 | 48.13 |
| | | | | **Tail Set** | | | | |
| max | 86.07 | 70.83 | 86.96 | 71.23 | 86.96 | 71.23 | 52.4 | 49.29 |
| min | 91.73 | 74.51 | 92.78 | 74.63 | 92.78 | 74.63 | 62.17 | 53.76 |
| count | 85.47 | 72.03 | 85.59 | 72.12 | 85.59 | 72.07 | 47.58 | 48.36 |
| sum | 81.35 | 46.31 | 83.47 | 46.87 | 83.47 | 46.68 | 44.0 | 33.66 |
| avg. | 50.0 | 58.25 | 50.0 | 61.23 | 50.0 | 59.31 | 35.94 | 60.86 |
| def. | 83.94 | 57.49 | 87.58 | 58.06 | 87.58 | 58.06 | 64.43 | 53.37 |
| com. | 76.92 | 56.23 | 76.92 | 55.98 | 76.92 | 55.98 | 43.75 | 37.23 |

Table 23: Comparison between Human and GPT-4 w.r.t. reasoning operations.

| Type | Human F1 | GPT-4 F1 | Human R1 | GPT-4 R1 | Human R2 | GPT-4 R2 | Human MET | GPT-4 MET |
|---|---|---|---|---|---|---|---|---|
| | | | | **Head Set** | | | | |
| explicit | 87.47 | 71.37 | 86.8 | 71.4 | 86.71 | 71.33 | 61.25 | 55.81 |
| implicit | 86.12 | 73.16 | 86.16 | 73.23 | 86.1 | 73.19 | 56.16 | 52.64 |
| | | | | **Tail Set** | | | | |
| explicit | 81.65 | 60.3 | 82.09 | 61.02 | 82.0 | 60.85 | 53.43 | 44.61 |
| implicit | 87.87 | 69.49 | 87.77 | 69.86 | 87.77 | 69.79 | 56.93 | 49.68 |

Table 24: Comparison between Human and GPT-4 w.r.t. type of temporal question

we introduced a warm-up period of 500 steps and applied weight decay at a rate of 0.01 during the optimization phase. We implemented logging and evaluation at every 100-step interval. The learning rate was designated at 2e-5, and a gradient accumulation process of 8 steps was used to optimize memory resources.

Our study further expanded to encompass zero-shot and few-shot experiments. This included, but was not limited to, chain-of-thought prompting on cutting-edge models like GPT-3.5 and GPT-4. We delved into various variants of Flan-T5 and T5 models, such as large L, XL, and XXL. A thorough analysis was undertaken to compare the performance of these models against human performance benchmarks.

## C   Models and Hyperparameters Details

In our research, we embarked on a series of training experiments utilizing several models such as BART, Flan-T5, and T5, each with base, large, and XL variants. Training was conducted over 1-3 epochs, incorporating input sequence lengths from 1024 to 4096 tokens. To foster efficient convergence,

| Type | Human F1 | GPT-4 F1 | Human R1 | GPT-4 R1 | Human R2 | GPT-4 R2 | Human MET | GPT-4 MET |
|---|---|---|---|---|---|---|---|---|
| **Head Set** | | | | | | | | |
| Boolean | 100.0 | 50.0 | 100.0 | 50.0 | 100.0 | 50.0 | 25.0 | 31.1 |
| Temporal | 86.74 | 72.4 | 86.98 | 72.46 | 86.91 | 72.43 | 61.16 | 60.22 |
| Count | 83.87 | 76.17 | 82.7 | 76.04 | 82.7 | 76.02 | 40.33 | 39.27 |
| Age | 84.91 | 58.21 | 84.91 | 58.21 | 84.91 | 58.17 | 30.99 | 30.97 |
| Money | 85.29 | 66.9 | 85.29 | 66.66 | 85.29 | 66.66 | 65.26 | 65.64 |
| Percentage | 62.5 | 45.19 | 62.5 | 45.72 | 62.5 | 45.72 | 36.42 | 21.33 |
| Ordinal | 66.67 | 50.0 | 66.67 | 50.0 | 66.67 | 50.0 | 28.33 | 0.0 |
| Place | 97.87 | 87.8 | 97.87 | 88.41 | 97.87 | 88.41 | 60.45 | 40.67 |
| Person | 92.62 | 68.9 | 92.49 | 68.81 | 92.49 | 68.81 | 62.32 | 38.98 |
| Organization | 91.79 | 84.24 | 91.79 | 84.23 | 90.92 | 84.1 | 70.5 | 39.68 |
| Product | 100.0 | 100.0 | 100.0 | 100.0 | 100.0 | 100.0 | 71.88 | 57.09 |
| Unknown | 86.51 | 72.16 | 85.82 | 72.62 | 85.82 | 72.43 | 55.2 | 46.58 |
| **Tail Set** | | | | | | | | |
| Boolean | 57.14 | 52.44 | 57.14 | 54.62 | 57.14 | 53.52 | 28.57 | 31.1 |
| Temporal | 85.52 | 67.43 | 85.61 | 67.63 | 85.55 | 67.52 | 63.4 | 60.22 |
| Count | 87.1 | 75.92 | 87.47 | 76.04 | 87.47 | 75.97 | 44.2 | 39.27 |
| Age | 91.96 | 60.57 | 91.93 | 60.75 | 91.93 | 60.67 | 45.95 | 30.97 |
| Money | 80.0 | 69.44 | 93.33 | 71.6 | 93.33 | 71.6 | 68.25 | 65.64 |
| Percentage | 23.33 | 35.71 | 22.22 | 37.25 | 22.22 | 37.25 | 8.33 | 21.33 |
| Ordinal | 0.0 | 0.0 | 0.0 | 0.0 | 0.0 | 0.0 | 0.0 | 0.0 |
| Place | 87.5 | 63.75 | 87.5 | 63.64 | 87.5 | 63.64 | 53.05 | 40.67 |
| Person | 75.09 | 38.69 | 75.09 | 39.07 | 75.09 | 38.84 | 65.4 | 38.98 |
| Organization | 95.35 | 61.46 | 95.35 | 66.6 | 95.35 | 66.6 | 61.21 | 39.68 |
| Product | 100.0 | 100.0 | 100.0 | 100.0 | 100.0 | 100.0 | 50.0 | 57.09 |
| Unknown | 86.38 | 62.89 | 85.48 | 63.6 | 85.48 | 63.48 | 61.06 | 46.58 |

Table 25: Comparison between Human and GPT-4 w.r.t. entity type

| Type | Human F1 | GPT-4 F1 | Human R1 | GPT-4 R1 | Human R2 | GPT-4 R2 | Human MET | GPT-4 MET |
|---|---|---|---|---|---|---|---|---|
| agency | 73.96 | 77.75 | 73.96 | 77.94 | 73.96 | 77.94 | 55.64 | 77.07 |
| aircraft | 89.58 | 76.61 | 85.42 | 76.32 | 85.42 | 76.32 | 57.7 | 58.45 |
| art | 100.0 | 93.33 | 100.0 | 93.33 | 100.0 | 93.33 | 62.59 | 59.4 |
| board game | 89.58 | 93.54 | 89.58 | 93.42 | 89.58 | 93.42 | 61.52 | 67.91 |
| book | 83.33 | 94.07 | 83.33 | 94.04 | 83.33 | 94.04 | 58.21 | 65.41 |
| character | 92.59 | 88.89 | 92.59 | 88.89 | 92.59 | 88.89 | 61.08 | 60.85 |
| church | 85.93 | 71.9 | 85.93 | 71.71 | 85.93 | 71.71 | 64.87 | 59.57 |
| company | 92.38 | 79.05 | 92.38 | 78.75 | 92.38 | 78.75 | 69.28 | 62.31 |
| concert | 88.89 | 89.42 | 88.89 | 89.38 | 88.89 | 89.38 | 69.56 | 68.52 |
| country | 81.48 | 78.74 | 81.48 | 78.63 | 81.48 | 78.63 | 54.93 | 75.45 |
| court | 87.62 | 86.19 | 87.62 | 86.17 | 87.62 | 86.17 | 68.68 | 70.67 |
| cricket team | 76.67 | 84.81 | 76.67 | 84.81 | 76.67 | 84.81 | 50.18 | 72.5 |
| disaster | 88.38 | 85.48 | 82.72 | 84.81 | 79.94 | 84.81 | 57.96 | 65.34 |
| event | 100.0 | 62.35 | 100.0 | 62.22 | 100.0 | 62.22 | 77.42 | 49.17 |
| fighter | 44.44 | 54.06 | 44.44 | 53.29 | 44.44 | 53.29 | 24.85 | 32.35 |
| finance | 77.42 | 54.21 | 77.42 | 53.97 | 77.42 | 53.97 | 52.96 | 40.33 |
| history | 82.96 | 50.48 | 82.96 | 51.09 | 82.96 | 51.09 | 58.15 | 36.8 |
| leaders | 85.93 | 61.59 | 85.93 | 61.65 | 85.93 | 61.49 | 63.95 | 51.06 |
| military conflicts | 88.57 | 65.13 | 81.43 | 64.29 | 81.43 | 64.29 | 46.77 | 41.83 |
| monument | 83.33 | 77.69 | 83.33 | 77.92 | 83.33 | 77.92 | 64.91 | 61.82 |
| movie | 72.92 | 68.06 | 72.92 | 68.55 | 72.92 | 68.55 | 47.58 | 46.46 |
| music | 100.0 | 68.18 | 100.0 | 68.18 | 100.0 | 68.18 | 72.74 | 48.68 |
| musician | 83.33 | 75.87 | 83.33 | 75.82 | 83.33 | 75.82 | 64.59 | 58.96 |
| unknown | 95.45 | 78.02 | 95.0 | 77.95 | 95.0 | 77.95 | 69.8 | 65.72 |
| national cricket team | 88.33 | 84.58 | 88.33 | 84.44 | 88.33 | 84.44 | 68.62 | 68.02 |
| national football team | 86.11 | 83.33 | 86.11 | 83.33 | 86.11 | 83.33 | 48.26 | 50.07 |
| nobel | 82.05 | 78.3 | 82.05 | 78.26 | 82.05 | 78.26 | 55.89 | 57.54 |
| office holders | 87.04 | 62.5 | 87.04 | 62.83 | 87.04 | 62.58 | 63.13 | 54.76 |
| person | 90.91 | 59.57 | 90.91 | 59.64 | 90.91 | 59.64 | 57.48 | 40.77 |
| rail line | 100.0 | 80.0 | 100.0 | 80.0 | 100.0 | 80.0 | 79.68 | 69.68 |
| railway | 93.33 | 87.69 | 93.33 | 87.62 | 93.33 | 87.62 | 77.3 | 75.52 |
| ships | 82.22 | 66.44 | 84.13 | 67.26 | 84.13 | 67.26 | 66.04 | 63.73 |
| song | 83.33 | 72.53 | 83.33 | 73.21 | 83.33 | 72.72 | 64.92 | 54.83 |
| space | 81.04 | 52.67 | 81.82 | 52.93 | 81.82 | 52.65 | 58.84 | 43.72 |
| sports | 86.66 | 71.34 | 86.81 | 71.45 | 86.73 | 71.41 | 54.93 | 51.29 |
| university | 100.0 | 76.1 | 88.89 | 74.77 | 88.89 | 74.27 | 54.51 | 53.94 |
| war | 86.47 | 61.99 | 86.47 | 62.51 | 86.47 | 62.51 | 56.02 | 41.8 |
| website | 100.0 | 100.0 | 100.0 | 100.0 | 100.0 | 100.0 | 79.17 | 79.17 |

Table 26: Comparison between Human and GPT-4 w.r.t. coarse-grained categories for Head Set

**Flan-T5** is an instruction fine-tuned derivative of the T5 language model, purposefully crafted to excel in a wide array of natural language processing tasks. These tasks include, among others, text generation, summarization, and translation. With the integration of instruction-based fine-tuning, Flan-T5 boosts its competency in handling zero-shot NLP tasks while also facilitating few-shot in-context learning. Thanks to its advanced encoder-decoder architecture and attention mech-

| Type | Human F1 | GPT-4 F1 | Human R1 | GPT-4 R1 | Human R2 | GPT-4 R2 | Human MET | GPT-4 MET |
|---|---|---|---|---|---|---|---|---|
| actor | 90.48 | 81.27 | 90.48 | 81.27 | 90.48 | 81.27 | 55.65 | 53.9 |
| agency | 73.96 | 77.75 | 73.96 | 77.94 | 73.96 | 77.94 | 55.64 | 77.07 |
| aircraft | 89.58 | 76.61 | 85.42 | 76.32 | 85.42 | 76.32 | 57.7 | 58.45 |
| album | 100.0 | 53.85 | 100.0 | 53.85 | 100.0 | 53.85 | 71.32 | 34.44 |
| athelete | 86.11 | 76.21 | 86.11 | 75.97 | 86.11 | 75.97 | 48.16 | 47.89 |
| badminton | 88.89 | 67.37 | 88.89 | 67.76 | 88.89 | 67.76 | 50.28 | 45.32 |
| baseball | 90.74 | 76.42 | 90.74 | 76.76 | 90.74 | 76.76 | 63.68 | 60.2 |
| basketball | 92.22 | 80.1 | 91.98 | 80.12 | 91.98 | 80.12 | 69.02 | 69.65 |
| board game | 96.3 | 97.62 | 96.3 | 97.49 | 96.3 | 97.49 | 66.17 | 72.07 |
| body builder | 88.75 | 70.63 | 88.75 | 70.5 | 88.75 | 70.5 | 64.75 | 58.99 |
| book | 83.33 | 94.07 | 83.33 | 94.04 | 83.33 | 94.04 | 58.21 | 65.41 |
| car driver | 76.19 | 67.92 | 76.19 | 67.99 | 76.19 | 67.85 | 47.68 | 53.96 |
| character | 92.59 | 88.89 | 92.59 | 88.89 | 92.59 | 88.89 | 61.08 | 60.85 |
| christian leader | 85.93 | 61.59 | 85.93 | 61.65 | 85.93 | 61.49 | 63.95 | 51.06 |
| church | 85.93 | 71.9 | 85.93 | 71.71 | 85.93 | 71.71 | 64.87 | 59.57 |
| civil war | 87.69 | 77.51 | 87.69 | 78.06 | 87.69 | 78.06 | 53.73 | 53.56 |
| company | 92.38 | 79.05 | 92.38 | 78.75 | 92.38 | 78.75 | 69.28 | 62.31 |
| concert | 88.89 | 89.42 | 88.89 | 89.38 | 88.89 | 89.38 | 69.56 | 68.52 |
| country | 81.48 | 78.74 | 81.48 | 78.63 | 81.48 | 78.63 | 54.93 | 75.45 |
| court | 87.62 | 86.19 | 87.62 | 86.17 | 87.62 | 86.17 | 68.68 | 70.67 |
| cricket team | 76.67 | 84.81 | 76.67 | 84.81 | 76.67 | 84.81 | 50.18 | 72.5 |
| curling | 87.5 | 64.95 | 87.5 | 64.92 | 87.5 | 64.92 | 49.46 | 45.44 |
| current war | 85.71 | 52.38 | 85.71 | 52.88 | 85.71 | 52.88 | 57.44 | 34.52 |
| earthquake | 88.38 | 85.48 | 82.72 | 84.81 | 79.94 | 84.81 | 57.96 | 65.34 |
| economy | 77.42 | 54.21 | 77.42 | 53.97 | 77.42 | 53.97 | 52.96 | 40.33 |
| emperor | 82.96 | 50.48 | 82.96 | 51.09 | 82.96 | 51.09 | 58.15 | 36.8 |
| empire | 95.45 | 78.02 | 95.0 | 77.95 | 95.0 | 77.95 | 69.8 | 65.72 |
| event | 100.0 | 62.35 | 100.0 | 62.22 | 100.0 | 62.22 | 77.42 | 49.17 |
| fighter | 44.44 | 54.06 | 44.44 | 53.29 | 44.44 | 53.29 | 24.85 | 32.35 |
| figure skating | 87.76 | 64.27 | 87.76 | 64.28 | 87.76 | 64.28 | 52.48 | 43.14 |
| footballer | 77.33 | 43.28 | 77.33 | 44.92 | 77.33 | 44.92 | 48.41 | 30.64 |
| game | 80.95 | 88.3 | 80.95 | 88.18 | 80.95 | 88.18 | 55.53 | 62.56 |
| golf | 81.48 | 51.85 | 81.48 | 51.85 | 81.48 | 51.85 | 52.41 | 39.38 |
| handball | 87.18 | 81.0 | 87.18 | 80.95 | 87.18 | 80.95 | 61.21 | 59.2 |
| ice hockey | 87.6 | 81.62 | 87.6 | 82.64 | 87.6 | 82.64 | 55.94 | 53.43 |
| lacrosse | 78.89 | 77.36 | 80.97 | 77.27 | 80.97 | 77.27 | 45.69 | 52.51 |
| launchpad | 56.67 | 52.26 | 56.67 | 52.05 | 56.67 | 52.05 | 43.19 | 34.6 |
| martial artist | 83.64 | 87.06 | 85.61 | 87.61 | 85.61 | 87.61 | 58.94 | 68.47 |
| military conflicts | 88.57 | 65.13 | 81.43 | 64.29 | 81.43 | 64.29 | 46.77 | 41.83 |
| monument | 83.33 | 77.69 | 83.33 | 77.92 | 83.33 | 77.92 | 64.91 | 61.82 |
| movie | 87.5 | 77.78 | 87.5 | 77.38 | 87.5 | 77.38 | 65.99 | 60.1 |
| music | 100.0 | 88.89 | 100.0 | 88.89 | 100.0 | 88.89 | 74.79 | 69.24 |
| musician | 83.33 | 75.87 | 83.33 | 75.82 | 83.33 | 75.82 | 64.59 | 58.96 |
| national cricket team | 88.33 | 84.58 | 88.33 | 84.44 | 88.33 | 84.44 | 68.62 | 68.02 |
| national football team | 86.11 | 83.33 | 86.11 | 83.33 | 86.11 | 83.33 | 48.26 | 50.07 |
| navy vessel | 82.22 | 66.44 | 84.13 | 67.26 | 84.13 | 67.26 | 66.04 | 63.73 |
| nba | 80.0 | 84.73 | 80.0 | 84.67 | 80.0 | 84.67 | 55.93 | 62.94 |
| nfl | 79.17 | 69.3 | 79.17 | 69.33 | 79.17 | 69.33 | 53.52 | 52.42 |
| nobel | 82.05 | 78.3 | 82.05 | 78.26 | 82.05 | 78.26 | 55.89 | 57.54 |
| office holders | 87.04 | 62.5 | 87.04 | 62.83 | 87.04 | 62.58 | 63.13 | 54.76 |
| painter | 100.0 | 93.33 | 100.0 | 93.33 | 100.0 | 93.33 | 62.59 | 59.4 |
| person | 94.87 | 44.03 | 94.87 | 44.15 | 94.87 | 44.15 | 61.06 | 31.02 |
| politician | 66.67 | 84.67 | 66.67 | 84.62 | 66.67 | 84.62 | 40.62 | 58.19 |
| racing | 76.92 | 66.49 | 76.92 | 66.3 | 76.92 | 66.3 | 48.9 | 47.75 |
| rail line | 100.0 | 80.0 | 100.0 | 80.0 | 100.0 | 80.0 | 79.68 | 69.68 |
| railway | 93.33 | 87.69 | 93.33 | 87.62 | 93.33 | 87.62 | 77.3 | 75.52 |
| rugby | 96.3 | 64.32 | 96.3 | 64.47 | 96.3 | 64.22 | 57.87 | 46.54 |
| sailor | 89.29 | 69.23 | 89.29 | 69.18 | 89.29 | 69.18 | 44.64 | 42.79 |
| scientist | 90.37 | 81.13 | 90.37 | 80.92 | 90.37 | 80.37 | 57.24 | 59.38 |
| show | 58.33 | 38.58 | 58.33 | 59.72 | 58.33 | 59.72 | 29.17 | 32.81 |
| skier | 94.44 | 79.28 | 94.44 | 79.23 | 94.44 | 79.23 | 56.94 | 48.37 |
| song | 83.33 | 72.53 | 83.33 | 73.21 | 83.33 | 72.72 | 64.92 | 54.83 |
| space probe | 81.22 | 53.09 | 82.7 | 53.25 | 82.7 | 52.71 | 66.67 | 48.72 |
| space program | 88.89 | 52.17 | 88.89 | 52.74 | 88.89 | 52.74 | 52.33 | 39.24 |
| stadium | 100.0 | 88.89 | 100.0 | 88.89 | 100.0 | 88.89 | 64.58 | 63.89 |
| swimming | 96.77 | 80.8 | 96.77 | 80.62 | 96.77 | 80.62 | 55.87 | 51.79 |
| table tennis | 100.0 | 73.81 | 100.0 | 73.71 | 100.0 | 73.71 | 62.5 | 47.22 |
| tennis | 90.0 | 61.03 | 90.0 | 61.23 | 90.0 | 61.23 | 58.27 | 42.66 |
| university | 100.0 | 76.1 | 88.89 | 74.77 | 88.89 | 74.27 | 54.51 | 53.94 |
| volleyball | 87.5 | 60.16 | 87.5 | 60.0 | 85.0 | 60.0 | 56.3 | 40.86 |
| website | 100.0 | 100.0 | 100.0 | 100.0 | 100.0 | 100.0 | 79.17 | 79.17 |
| wrestling | 81.82 | 66.18 | 81.82 | 66.06 | 81.82 | 66.06 | 55.03 | 55.25 |

Table 27: Comparison between Human and GPT-4 w.r.t. fine-grained categories for Head Set

| Type | Human F1 | GPT-4 F1 | Human R1 | GPT-4 R1 | Human R2 | GPT-4 R2 | Human MET | GPT-4 MET |
|---|---|---|---|---|---|---|---|---|
| aircraft | 83.33 | 70.8 | 83.33 | 70.52 | 83.33 | 70.52 | 64.27 | 55.87 |
| army | 77.14 | 55.91 | 77.68 | 56.54 | 77.68 | 56.32 | 55.05 | 47.01 |
| disaster | 74.6 | 44.12 | 74.6 | 44.53 | 74.6 | 44.32 | 49.08 | 29.99 |
| diseases | 67.94 | 59.37 | 70.46 | 59.9 | 70.46 | 59.6 | 42.98 | 41.08 |
| holiday | 76.29 | 57.37 | 77.01 | 57.71 | 77.01 | 57.29 | 48.49 | 44.26 |
| organization | 91.67 | 71.35 | 91.67 | 70.59 | 91.67 | 70.59 | 53.12 | 70.0 |
| party | 86.38 | 72.89 | 86.38 | 73.33 | 86.38 | 73.33 | 62.83 | 57.61 |
| planet | 83.33 | 83.33 | 83.33 | 83.33 | 83.33 | 83.33 | 56.25 | 56.25 |
| ships | 84.69 | 67.61 | 84.69 | 68.24 | 84.69 | 68.24 | 61.13 | 51.91 |
| sports | 91.02 | 71.96 | 91.03 | 72.46 | 90.98 | 72.42 | 55.32 | 48.08 |
| time zone | 87.22 | 61.85 | 79.33 | 62.03 | 79.33 | 62.03 | 64.51 | 55.11 |
| war | 87.94 | 59.5 | 87.62 | 60.01 | 87.62 | 59.93 | 66.19 | 49.49 |

Table 28: Comparison between Human and GPT-4 w.r.t. coarse-grained categories for Tail Set

| Type | Tail Domain | | | | | | | |
|---|---|---|---|---|---|---|---|---|
| | Human F1 | GPT-4 F1 | Human R1 | GPT-4 R1 | Human R2 | GPT-4 R2 | Human MET | GPT-4 MET |
| army | 77.14 | 55.91 | 77.68 | 56.54 | 77.68 | 56.32 | 55.05 | 47.01 |
| boxing | 88.89 | 67.97 | 88.89 | 67.84 | 88.89 | 67.84 | 59.52 | 50.04 |
| cricket | 89.29 | 70.29 | 89.29 | 70.2 | 89.29 | 70.2 | 53.65 | 44.37 |
| cycling | 95.16 | 62.79 | 95.06 | 65.51 | 95.06 | 65.51 | 55.53 | 41.64 |
| cyclone | 74.6 | 44.12 | 74.6 | 44.53 | 74.6 | 44.32 | 49.08 | 29.99 |
| disease | 67.94 | 59.37 | 70.46 | 59.9 | 70.46 | 59.6 | 42.98 | 41.08 |
| f1 | 93.89 | 75.9 | 93.89 | 75.94 | 93.89 | 75.94 | 55.96 | 50.46 |
| hockey | 92.27 | 72.27 | 92.27 | 72.23 | 92.27 | 72.23 | 54.05 | 44.19 |
| holiday | 76.29 | 57.37 | 77.01 | 57.71 | 77.01 | 57.29 | 48.49 | 44.26 |
| orbitor | 83.33 | 70.8 | 83.33 | 70.52 | 83.33 | 70.52 | 64.27 | 55.87 |
| planet | 83.33 | 83.33 | 83.33 | 83.33 | 83.33 | 83.33 | 56.25 | 56.25 |
| political party | 86.38 | 72.89 | 86.38 | 73.33 | 86.38 | 73.33 | 62.83 | 57.61 |
| proxy war | 87.94 | 59.5 | 87.62 | 60.01 | 87.62 | 59.93 | 66.19 | 49.49 |
| ship | 84.69 | 67.61 | 84.69 | 68.24 | 84.69 | 68.24 | 61.13 | 51.91 |
| sports event | 86.35 | 82.93 | 86.78 | 83.02 | 86.28 | 82.78 | 58.24 | 59.77 |
| squash | 87.5 | 85.11 | 87.5 | 85.11 | 87.5 | 85.11 | 50.22 | 50.49 |
| sumo | 88.65 | 64.12 | 88.61 | 64.18 | 88.61 | 64.07 | 58.57 | 51.88 |
| terrorist orgnization | 91.67 | 71.35 | 91.67 | 70.59 | 91.67 | 70.59 | 53.12 | 70.0 |
| time zone | 87.22 | 61.85 | 79.33 | 62.03 | 79.33 | 62.03 | 64.51 | 55.11 |

Table 29: Comparison between Human and GPT-4 w.r.t. fine-grained categories for Tail Set

anisms, Flan-T5 efficiently leverages contextual information to generate high-quality output. This capability promotes opportunities for enhanced performance and adaptability across various language processing applications.

Our study involved fine-tuning the Flan-T5 model across its different variants: Flan-T5-Base, Flan-T5-Large, and Flan-T5-XL. Alongside this, we also carried out zero-shot, few-shot (with and without chain of thought prompting) experiments on Flan-T5-Large, Flan-T5-XL, and Flan-T5-XXL.

**BART (Bidirectional and AutoRegressive Transformers)** is a robust sequence-to-sequence model architecture widely adopted in various natural language processing tasks. By fusing bidirectional and autoregressive training objectives, BART is capable of exploiting the context of both the input and target sequences. Given that BART features an autoregressive decoder, it can be directly fine-tuned for sequence generation tasks, such as abstractive question answering.

We fine-tuned the BART-Large model, consisting of 12 encoder-decoder layers with 440 million parameters, and BART-Base model, comprising 6 encoder-decoder layers and 140 million parameters. The performance analysis and outcomes of these fine-tuned models can be found in the main paper.

**T5 (Text-To-Text Transfer Transformer)** is a versatile language model architecture. Based on the transformer model, T5 is equipped to handle various natural language processing tasks. By leveraging a "text-to-text" training approach, T5 learns to transform input text into target text, thus enabling it to manage a wide variety of tasks. These tasks include text classification, summarization, translation, and question answering. The T5 model

incorporates an encoder-decoder structure with several layers of self-attention mechanisms and uses a shared vocabulary and tokenization scheme, thereby ensuring a consistent representation and efficient processing of text data.

We carried out fine-tuning on different variants of the T5 model: T5-Base, T5-Large, and T5-XL. We also conducted zero-shot experiments on T5-Large, T5-XL, and T5-XXL.

**GPT-3.5-turbo and GPT-4** are the latest developments in the distinguished GPT series of language models. GPT-3.5 Turbo is an enhanced variant of GPT-3, boasting approximately 154 billion parameters and demonstrating superior language processing capabilities. It is particularly adept at text generation, comprehension, summarization, among other tasks. Conversely, GPT-4 signifies the next step in language modeling with an expected model size of around 1 trillion parameters and improved language understanding and generation capacities. These models rely on vast pretraining data for superior generalization and exhibit excellent performance in both zero-shot and few-shot learning scenarios.

We conducted a suite of experiments on GPT-3.5 Turbo and GPT-4, focusing on zero-shot and few-shot learning scenarios, examining their performance with and without reasoning capabilities.

**Table Representation.** Firstly, each table is transformed from HTML into a JSON representation, containing subheadings, rows with keys and their respective values, as well as the table title and category as distinct keys. We employed a linearization process akin to INFOTABS (Gupta et al., 2020), using delimiters such as "tab" or ":" to separate keys, and "newline" or ";" for rows. Subsections are partitioned by double "new lines" or "#". For instance, in Table 1, the representation is: Title: Petya Nedelcheva # Personal Information # Country: Bulgaria; Born: July 30, 1983 (age 38), and soon.

# D Crowdsourcing Details

To construct TEMPTABQA, we divided the task into 80 batches, each consisting of three question-answer pair generations per HIT[6]. We assigned

---

[6]A Human Intelligence Task, or HIT, is a question that needs an answer. A HIT represents a single, self-contained, virtual task that a Worker can work on, submit an answer, and collect a reward for completing. HITs are created by Requester customers in order to be completed by Worker customers.

| Dataset | Number | Gold/total |
|---------|--------|-----------|
| Dev | 3 | 82/91 |
| | 4 | 3/3 |
| | 5 | 1/2 |
| | Overall | 148/166 |
| Head | 3 | 898/1033 |
| | 4 | 20/32 |
| | 5 | 23/32 |
| | Overall | 1627/1821 |
| Tail | 3 | 386/468 |
| | 4 | 26/33 |
| | 5 | 39/49 |
| | Overall | 999/1155 |

Table 30: Exact agreement between annotators

each HIT to three distinct annotators, resulting in an average of 9 QA pairs generated per table. The wage for each HIT, which involved generating three question-answer pairs for a given table, was set at 0.75 cents based on the average completion time observed during three pilot studies.

All our annotators were proficient English speakers from countries where English is spoken. They possessed master-level qualifications and maintained a HIT acceptance rate of 95% and above. We occasionally rewarded frequent and exceptional annotators with a bonus of 3 times the cost of the HIT. To ensure task quality, we implemented temporary blocking and rewarding mechanisms for annotators.

For verification purposes, each HIT required answering three questions and providing a brief explanation. An annotator received 0.15 cents per HIT for this task. If consensus was not reached among the initial three annotators, we reassigned the HIT to another set of three annotators. Here two we start with three pilot study to decide the cost of the annotation. Notably, we observed that the top 50 annotators were responsible for annotating approximately 90% of the dataset. This observation aligns with other crowdsourced data annotation projects such as SNLI and MultiNLI.

**Validation Details.** We employ straightforward pre-processing scripts to remove non-temporal and basic extractive questions from the training set prior to fine-tuning. For the test and development sets, we enforce rigorous quality control by manually reviewing each Table QA, with input from three experts who are NLP researchers. This process follows the initial automated script-based filtering and is aimed at ensuring high-quality complex temporal questions. Additionally, we address answer

units and correct spelling errors during our quality filtering. We prioritize questions involving intricate temporal reasoning and abstract concepts while filtering out questions with answers directly present in the question or associated tables. Questions that required external knowledge beyond common sense are also filtered.

Our annotators were directed to offer concise and pertinent answers. While the majority adhered to the instructions, a few instances deviated, leading to occasional ambiguities. These ambiguities typically emerged when multiple answer forms conveyed the same meaning but in different units or formats, such as '365 days' '12 months' or 'one year.' We ensured our assessment script didn't penalize models or human verifiers for unit or format conversion issues. We established regex rules that encompassed various forms, and these were further validated through human verification across numerous samples. Figure 30 shows the exact agreement between across several annotators.

# E More Examples from TEMPTABQA

Figures 2, 3, 4, 5, 6 show some examples of tabular question answers from TEMPTABQA.

Answering these questions demands from language models an understanding of temporal relationships to correctly connect time frames to pertinent events, as well as numerical reasoning to perform calculations, comparisons, and quantitative analyses based on temporal data. This includes both basic arithmetic and complex numerical reasoning like identifying trends or evaluating numerical changes over time.

These questions present challenges for language models due to the multi-faceted nature of the information required to answer them. First, they demand a deep understanding of temporal relationships, encompassing the ability to interpret and analyze time frames accurately. The language model must effectively connect these time frames to specific events, albums, or other relevant entities mentioned in the context.

Furthermore, numerical reasoning plays a crucial role in successfully addressing these questions. The language model needs to perform calculations, comparisons, and quantitative analysis based on temporal data to arrive at the correct answers. This entails not only basic arithmetic operations but also more sophisticated numerical reasoning, such as identifying trends, computing durations, or evaluat-

| Dataset | Temporal | Entity | Domain | Task | Hybrid | Numerical | Synthetic | Abstractive |
|---------|----------|--------|--------|------|--------|-----------|-----------|-------------|
| INFOTABS | ✗ | ✓ | Generic | NLI | ✗ | ✓ | Human | ✓ |
| TABFACT | ✗ | ✗ | Generic | NLI | ✗ | ✓ | Human | ✓ |
| LOGICQA | ✗ | ✗ | Generic | QA | ✗ | ✓ | Machine | ✓ |
| FETAQA | ✗ | ✗ | Generic | QA | ✗ | ✓ | Machine | ✗ |
| FINQA | ✗ | ✗ | Finance | QA | ✓ | ✓ | Human | ✗ |
| TATQA | ✗ | ✗ | Finance | QA | ✓ | ✓ | Human | ✗ |
| HYBRIDQA | ✗ | ✗ | Generic | QA | ✓ | ✓ | Human | ✗ |
| WIKITABLEQA | ✗ | ✗ | Generic | QA | ✗ | ✗ | Human | ✗ |
| SQUALL | ✗ | ✗ | Generic | QA | ✗ | ✗ | Human | ✓ |
| WIKISQL | ✗ | ✗ | Generic | QA | ✗ | ✗ | Human | ✓ |
| SQA | ✗ | ✗ | Generic | QA | ✗ | ✓ | Human | ✓ |
| TEMPTABQA | ✓ | ✓ | Generic | QA | ✗ | ✓ | Human | ✓ |

Table 31: Comparison of TEMPTABQA with existing standard Tabular Datasets.

| Statistic | WIKITABLEQA | HYBRIDQA | SQA | FINQA | TABFACT | FETAQA | SQUALL |
|-----------|-------------|----------|-----|-------|---------|--------|--------|
| # temporal | 7807 | 33713 | 4634 | 6774 | 11219 | 5978 | 3876 |
| temporal (%) | 35.46 | 48.43 | 26.40 | 82.48 | 37.69 | 57.87 | 34.37 |
| **Question Types** | | | | | | | |
| explicit | 5830 | 28441 | 2899 | 6719 | 10797 | 5657 | 2901 |
| implicit | 1977 | 5272 | 1735 | 55 | 422 | 321 | 975 |
| ordinal | 1433 | 5267 | 276 | 149 | 107 | 510 | 754 |
| **Temporal Interval** | | | | | | | |
| before | 1006 | 1156 | 87 | 70 | 173 | 109 | 539 |
| after | 524 | 893 | 55 | 159 | 41 | 83 | 259 |
| duration | 1249 | 2520 | 391 | 1070 | 2218 | 562 | 567 |
| yes/no | 86 | 6 | 10 | 103 | 9 | 91 | 65 |
| temporal | 1783 | 8588 | 1845 | 52 | 1605 | 4681 | 526 |
| **Operation Involved** | | | | | | | |
| max | 531 | 1735 | 185 | 78 | 119 | 115 | 274 |
| min | 597 | 1272 | 198 | 344 | 219 | 352 | 294 |
| count | 2313 | 5789 | 273 | 223 | 597 | 706 | 1109 |
| sum | 619 | 965 | 128 | 1960 | 1954 | 254 | 316 |
| difference | 249 | 225 | 23 | 2390 | 4149 | 60 | 127 |
| average | 124 | 391 | 33 | 1788 | 2633 | 38 | 41 |
| comparison | 470 | 697 | 184 | 126 | 834 | 38 | 222 |

Table 32: Statistics of temporal question present in the existing tabular datasets. Most of the question are explicit and involve only numerical reasoning.

ing numerical changes over time.

## F  Further Discussion

**Key Findings.** Based on our experimental analysis in §4, we conclude that even state-of-the-art large language models like GPT-4 struggle with temporal question answering on entity-centric tables within TEMPTABQA, despite humans' high performance. Fine-tuning and few-shot learning techniques have a positive impact on the model's performance. The model encounters more difficulties in the tail domain, comprising rare occurrences, compared to the head domain with more frequent instances. Techniques involving step-by-step explanations, such as chain of thought prompting, further enhance the model's performance.

In our breakdown in §5, we discovered inconsistent performance of the model and humans across various question types, answer entity types, reasoning operations, answer positions, and table domains. Both the model and humans demonstrate varying levels of proficiency across different categories. The analysis helps identify weaknesses and areas for improvement in future temporal reasoning models on semi-structured tabular data.

**Semi-structured Tables.** Semi-structured data lies in a realm between raw, unstructured text and rigidly structured content such as Knowledge Grpah. This data landscape, where structured frameworks interweave with free-form text, spans the gamut from extensive verbosity like web pages, to succinct instances such as fact sheets, information tables, and technical specifications. Unlike databases, this type of data isn't uniformly structured; it can be a heterogeneous assortment without preset schemas. Adding to the complexity, explana-

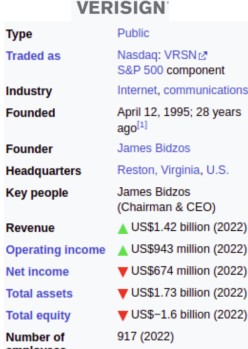

Q1: What was Verisign's operating income the year the number of employees reached 904? A1: 886 million
Q2: Is net income of Verisign, Inc. was reduced in 2021 compared to 2020? A2: Yes
Q3: How long did it take for Verisign to reach over 900 employees? A3: 26 years
Q4: Who was the CEO in 1995? A4: James Bidzos

Figure 2: A semi-structured table (source: Wikipedia) along with accompanying temporal questions and their respective answers form TEMPTABQA.

tory text that imparts context isn't always at hand. Nonetheless, we frequently deduce insights from such diverse and incomplete data, bridging information gaps based on our expectations about relationships within.

**Reasoning Requirements.** Navigating semi-structured information necessitates a broad range of reasoning skills. We're tasked with comprehending a makeshift layout composed of elements like text snippets, form fields, or even sub structured components like lists. Querying this data calls for various levels of inference, ranging from straightforward lookups e.g. in Figure 1 querying Petya born place, to lexical deductions, such as understanding in same table single (WS) and double game (WD) format, junior championship vs. senior events of badminton, to grasping the nature of content within cells, the structure of the various events, the tournament names, tournament years and places, the total and specific medals tally, and the tournament types. Additionally, we might find ourselves aggregating insights across multiple rows, such as understanding that Dressage is a non-contact sport in which both genders compete, or even conducting intricate reasoning that melds temporal details with general knowledge.

**Similarity with Knowledge Graph.** However, it's important to note that Infoboxes exhibit a high degree of similarity with standard knowledge bases, particularly when compared to Wikidata. Wikidata generally surpasses Infoboxes in terms of comprehensiveness. When contrasting Wikidata with the

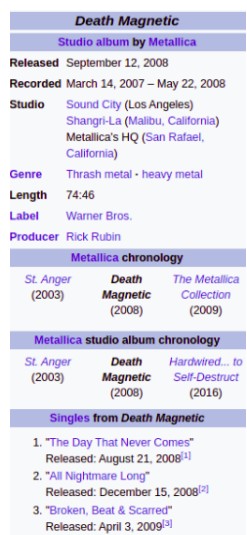

Q1: How many singles were released in the same year Death Magnetic was released? A1: 2
Q2: How many months did it take for Metallica to record Death Magnetic? A2: 14 Months
Q3: How many years after Death Magnetic did Metallica record another studio album? A3: 8 years
Q4: How many singles were released on the Death Magnetic album in 2008-2009? A4: 3

Figure 3: A semi-structured table (source: Wikipedia) along with accompanying temporal questions and their respective answers form TEMPTABQA.

Infobox style, we observe significant distinctions in how information is structured. Wikidata adopts a more organized and structured approach, resembling a knowledge graph. For example, when dealing with a person's birth details, Wikidata neatly separates the information into distinct categories like "birth date," "birth place," and "birth name." In semi-structured Infoboxes , on the other hand, these details are often combined under a single heading, such as "Born." Furthermore, there is a noticeable contrast in how relationships are presented. In Wikidata, relationships are systematically categorized. For instance, instead of using a generic "spouse" label, Wikidata provides separate entries for "husband" and "wife," resulting in a more precise representation. In contrast, an Infobox might consolidate such information under a single "spouse" entry without specifying the gender.

**From a Temporal Perspective** Temporal details find distinct treatment as well. Wikidata distinctly separates "start date" and "end date," yielding precise timeline information. This stands in contrast to Infoboxes , where these details could be condensed into single terms like "service," potentially necessitating further interpretation. Wikidata's penchant for hierarchy is evident in how complex terms are

| Miloš Obrenović I | |
|---|---|
| **Reign** | 23 December 1858 – 26 September 1860 |
| **Predecessor** | Alexander Karađorđević |
| **Successor** | Mihailo III (Obrenović) |
| **Prince of Serbia** | |
| **Reign** | 6 November 1817 – 25 June 1839 |
| **Predecessor** | Himself (As Grand Vožd of Serbia) |
| **Successor** | Milan II |
| **Grand Vožd of Serbia** | |
| **Reign** | 23 April 1815 – 6 November 1817 |
| **Predecessor** | Karađorđe |
| **Successor** | Himself (as Prince of Serbia) |
| **Born** | 18 March 1780 or more probably 1783 Gornja Dobrinja near Požega, Ottoman Empire (now Serbia) |
| **Died** | 26 September 1860 (aged 77 or 80) Belgrade, Serbia, Ottoman Empire |
| **Burial** | St. Mark's Church, Belgrade, Serbia |
| **Consort** | Ljubica Vukomanović |
| **Issue** | Princess Petria Princess Elisabeth Prince Milan Obrenović Prince Mihailo Obrenović Princess Maria Prince Todor Prince Gabriel |
| **House** | Obrenović |
| **Father** | Teodor Mihailović |
| **Mother** | Višnja Urošević |
| **Religion** | Serbian Orthodox |

Q1: Who was the Prince of Serbia in 1857 before Miloš Obrenović I? A1: Alexander Karađorđević

Q2: How many years before his death did Miloš Obrenović I begin his second reign as the Prince of Serbia? A2: 2 years

Q3: How long after Miloš Obrenović 's reign as Grand Vožd of Serbia ended did his second reign as Prince of Serbia begin? A3: 41 years

Q4: How many years elapsed between reigns of Prince of Serbia for Miloš Obrenović I? A4: 19 years

Q5: Who was the Prince of Serbia in 1840? A: Milan II

Figure 4: A semi-structured table (source: Wikipedia) along with accompanying temporal questions and their respective answers form TEMPTABQA.

broken down. For instance, a "government official" could be subcategorized as "president," "prime minister," and more. In contrast, Infoboxes might lack this hierarchical clarity, opting for more generalized terms. Granular attributes shine in Wikidata, with individual specifications for attributes like "awards," enabling a detailed breakdown of accolades. Conversely, Infoboxes could consolidate these attributes, obscuring the specifics of received awards. When it comes to event descriptions, Wikidata adopts a distinction between "start time" and "end time," leading to lucid event elucidations. In Infoboxes , these might be captured by a singular term, potentially devoid of temporal context. Lastly, Wikidata's categorization of properties imparts a structured approach to data.

| Aude Gemma Billard | |
|---|---|
| **Born** | 1971 (age 51–52) Lausanne, Switzerland |
| **Nationality** | Swiss |
| **Alma mater** | B.S. and M.S. École Polytechnique Fédérale de Lausanne (EPFL), M.S. and Ph.D. University of Edinburgh |
| **Known for** | Applying machine learning to robotics to improve learning and task performance |
| **Awards** | 2016 Nominated as Member of SATW, Swiss Academy of Engineering Sciences, 2016 Nominated for Outstanding Women in Academics SNSF, 2015 King-Sun Fu Best Transactions Paper Award, IEEE & Robotics and Automation Society, 2003 The Outstanding Young Person in Science and Innovation, Junior Chamber of Commerce, 2002 SNF Professeur Boursier, Career Award from Swiss National Science Foundation, 2001 Innovative Teaching Grant - Intel Corporation, 1999 Fellowship Medicus Foundation, 1996-97 Scholarship, Swiss National Science Foundation |
| **Scientific career** | |
| **Fields** | Machine learning, robotics, physics |

Q1: What award did Billard receive the same year she was Nominated as Member of SATW? A1: Nominated for Outstanding Women in Academics SNSF

Q2: What award did Aude Gemma Billard get when she was 44 years old? A2: King-Sun Fu Best Transactions Paper Award

Q3: How many years are between Billard's award for The Outstanding Young Person in Science and Innovation and Innovative Teaching Grant? A3: 2 years

Q4: How many years ago did Aude Gemma Billard was got award for Fellowship Medicus Foundation? A4: 23 Years ago (1999)

Figure 5: A semi-structured table (source: Wikipedia) along with accompanying temporal questions and their respective answers form TEMPTABQA.

In contrast, Infoboxes may not adhere to a similar systematic categorization, potentially leading to ambiguity. Collectively, these instances highlight the structured nature of Wikidata in contrast to the more succinct, semi-structured implicit knowledge of Infoboxes .

**Entity Table $\xrightarrow{conversion}$ Knowledge Graph.** As Infoboxes are highly structured (compared to Web tables), we could translate them to Wikidata and apply existing datasets and algorithms. Despite, this approach holds some promise, it's worth noting that transforming them into a clean and fully structured Wikidata format is in itself a challenging task, as highlighted earlier. Nevertheless, it presents an interesting opportunity to explore the capabilities of state-of-the-art language models in achieving this conversion. However, it's important to acknowledge that tables not found on Wikipedia, such as those containing e-commerce attribute values, research grants, medical reports, financial company data, etc., pose their own challenges when it comes to transitioning them into structured knowledge formats like Wikidata.

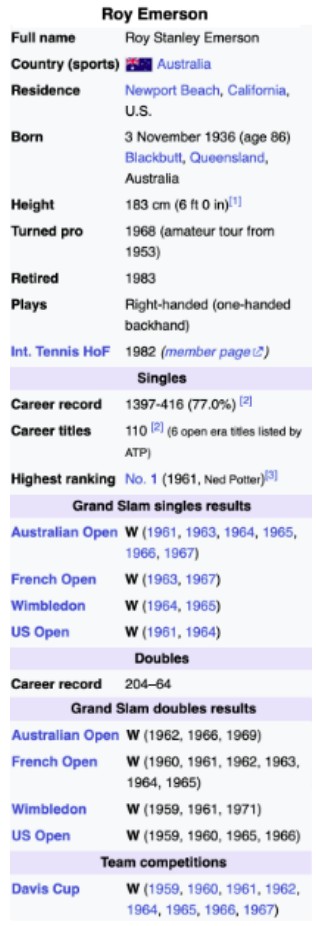

Q1: When was the most recent time that Roy Emerson won the Australian Open? A1: 1969

Q2: How long did Roy Emerson play in the amateur league before going professional? A2: 15 years

Q3: How old was Roy Emerson when he retired from playing professionally? A3: 47 years old

Q4: What year was Roy Emerson inducted into the International Tennis Hall of Fame? A4: 1982

Q5: Which Grand Slam tournament did Emerson win the 2nd time he won the Davis Cup team competition? A5: French Open

Figure 6: A semi-structured table (source: Wikipedia) along with accompanying temporal questions and their respective answers form TEMPTABQA.

## G Table Representation for LLMs

We experimented with three prompts, each featuring detailed instructions similar to those given to human verifiers. These prompts were based on three distinct table representations using different delimiters. Our selection process involved choosing the prompt that yielded the best performance. We present the table input in a linear format, akin to the approach adopted in TABFACT (Chen et al., 2020b) and INFOTABS (Gupta et al., 2020). Here, we employ a distinctive denominator token to demarcate rows using ";" and columns using ":". We also explored alternative delimiters such as "|" and "#" as well, the performance was similar.

We also experimented with an approach involving attempted table-to-paragraph conversion, but it caused models to include unwanted external information. LLM parametric knowledge lead to out of table unwanted hallucination in the paragraph. The performance variation across these representations were marginal $<1\%$ in the F1-score, and $<0.75\%$ in the exact match.

## H Future Directions: Other Modeling Techniques

Based on our observations and discussions, we have identified several promising future directions for enhancing models performance on TEMPT-ABQA:

1. **LLM Pre-trained with Temporal Knowledge:** Explore techniques incorporating temporal aspects during pre-training for masked language models (e.g., Dhingra et al. (2022); Iv et al. (2022)). Assess their performance in temporal tabular tasks using auxiliary tasks from temporal question-answering datasets in open domains (Jia et al., 2021b), cloze-form, or event-centric settings (Dhingra et al., 2022; Chen et al., 2021a; Ning et al., 2018; Wen et al., 2021).

2. **Temporal-Aligned Models for Entity-Centric Tabular Data:** Utilize temporally tuned language models (e.g., TEQUILA, EXAQT, OTR-QA, TempoQR) on temporal knowledge-based question-answering datasets (e.g., CRONQUESTIONS (Saxena et al., 2021), TEMPQA-WD (Neelam et al., 2022)) for answering questions related to temporal events (Jia et al., 2018a,b; Shang et al., 2021; Mavromatis et al., 2021; Saxena et al., 2021; Neelam et al., 2022).

3. **Integrating External Temporal Knowledge:** Incorporate knowledge base question-answering datasets like CRONKBQA(Saxena et al., 2021), nto LLM models during pre-training (e.g., ERNIE (Zhang et al., 2019), WKLM, KECP, ERICA, DKPLM) or structural adaptation (e.g., ERNIE-THU, Know-Bert, EaE, JAKET). Explore the use of non-entity temporal relations (e.g., ERICA, KEPLER, DKPLM, KP-PLM) through pre-training objectives or structural adaptation methods (e.g., FaE, K-adapter, KB-adapter,

KLMO, KERM, JointLK, GreaseLM, JAKET, KnowPrompt, OntoPrompt) as described in detail in (Hu et al., 2023).

4. **Fine-Tuning on Other Temporal Knowledge:** Investigate benefits of training on synthetic and counterfactual temporal data (implicit knowledge addition) to enhance model performance, similar to AUTOTNLI (Kumar et al., 2022) and (Eisenschlos et al., 2020). Consider using simple temporal data from unstructured text sources like Time-Sensitive-QA (Chen et al., 2021c) and CogCompTime (Ning et al., 2018), or structured text datasets like TempQA-WD, CronQUESTIONS, and TempQuestions (Saxena et al., 2021; Neelam et al., 2022; Jia et al., 2018b; Shang et al., 2021), which feature question-answering over knowledge graph embeddings with temporal links.