# OpenReview forum: "TempTabQA: Temporal Question Answering for Semi-Structured Tables"
_EMNLP/2023/Conference — EMNLP 2023 Main_

### Official Review · Reviewer_XpmN · 2023-08-03

**Typos Grammar Style And Presentation Improvements:** Nothing
**Soundness:** 3

**Excitement:**

4: Strong: This paper deepens the understanding of some phenomenon or lowers the barriers to an existing research direction.

**Missing References:**

Nothing

**Paper Topic And Main Contributions:**

In this paper, the authors introduce a new dataset for temporal question answering over Wikipedia's InfoBox. They describe how the dataset is created, give statistics, and run baselines with large language models. Overall, it is a valuable resource for the community, even if some points in the paper, like the evaluation of the dataset and the baselines, can be improved.

**Questions For The Authors:**

Q1: Can you adapt state-of-the-art algorithms to your dataset? Which ones and how?

Q2: How is the input of the models formatted?

Q3: More generally, answer the remarks above.

**Reasons To Accept:**

S1: The dataset is a valuable resource for the community. It is created by humans and comprises a reasonable amount of varied data.

S2: The authors provide an excellent statistical analysis of the diversity of the dataset.

S3: The authors extract interesting insights from the baseline analysis.

S4: The paper is well-written and easy to follow.

**Reasons To Reject:**

W1: The comparison with the previous work is very slim and briefly mentioned at the end of the paper. For the dataset, how does it compare to other datasets from a statistical point of view (number of questions, diversity)? Table 31 is in the annex and lacks information. Besides, the authors quickly evade the question of the knowledge graph question answering. However, infoboxes are highly close to standard knowledge bases, particularly Wikidata (they are automatic conversions between Wikidata and Infoboxes, and Wikidata is generally much more complete). As infoboxes are highly structured (compared to Web tables), we could translate them to Wikidata and apply existing datasets and algorithms. For the baselines, only raw language models are considered, whereas more advanced solutions could have been adopted (also briefly mentioned at the end). Although it is not the core goal of the paper, it could help future works.

W2: The authors do not explicitly evaluate the quality of the dataset. In section 3.3, they give good criteria for the evaluation (can we answer only with the table? Is the answer ambiguous? Is it an opinion?). Still, they are not used for the evaluation of the dataset. The human accuracy is about 86% for the exact match. Where do the errors come from (formulation? hard questions? misunderstanding?)

W3: More detail on the baseline would be nice. How is the input of the model formatted? This is quite an important point due to the structure of infoboxes. Besides, I wish the authors would have tried to adapt state-of-the-art architecture for (temporal) question answering.

W4: Some analyses in section 5 are sometimes statistically irrelevant. I wish there were p-values. Some sample sizes are too small to draw conclusions (1, 16, 21, 22).


Other points:
O0: The authors could have provided the dataset and code for the baselines on an anonymous link. It is currently impossible to check what the authors said.

O1: The paper's title is strangely formatted: TEMPTABQA is composed of several words, making it hard to copy-paste it (and parse for tools like Google Scholar).

O2: Would it be helpful to include synonym answers for a better exact match score? For example: "How many years..." could lead to "3", "3 years", "three," or "three years." Same thing for dates. Did you have a constraint that the Turks should or should not put the unit?

O3: Line 42-44: Research on the *temporal aspect* was limited.

O4: Can a question have several correct answers?

O5: Section 5: cite the papers for the models you use.

O6: It is a pity the authors were not able to include bigger models for the finetuned models, like LLaMa or Bloom. I understand the technical difficulties, though. Algorithms like PEFT can help.

O7: In the table, put the best results in bold and the second best underlined.

O8: The abbreviations F1, EM, R1, R2, MET should be introduced somewhere.

O9: Line 354: Is it surprising?

O10: Paragraph line 366: The difference between head and tail is slim for finetuned models. Why? This should be reported.

O11: Section 6 is not helpful. It is a kind of conclusion in the middle of the paper that only repeats the previous section. I do not see how the future directions are related to the previous discussions. I would put the section at the end.

O12: What does HIT mean in Section D?

**Reproducibility:**

5: Could easily reproduce the results.

**Reviewer Confidence:**

4: Quite sure. I tried to check the important points carefully. It's unlikely, though conceivable, that I missed something that should affect my ratings.

---

> ### Author Rebuttal · Authors · 2023-08-29
>
> Thank you for taking the time to read our article and provide valuable feedback.  We appreciate your thoughts and concerns and address them below:
>
> Reasons to Reject
>
> W1: The comparison with the previous work is very slim and briefly mentioned at the end of the paper. For the dataset, how does it compare to other datasets from a statistical point of view (number of questions, diversity)? Table 31 is in the annex and lacks information.  Besides, the authors quickly evade the question of the knowledge graph question answering.
>
> The prior work mostly dealt with non-temporal questions on database style tables (WebTableQA, SQUALL, FinQA, TaTQA, and others mentioned in the Related work Section 7). Others include temporal questions on either unstructured data (Time-sensitive QA and others) or  fully  structured, clean data, such as knowledge graphs (CronsQuestions, TempQuestions and others). Our work focuses on entity centric tables such as Wikipedia infoboxes. Entity centric tables such as Wikipedia infoboxes are very different (semi-structured) from both unstructured and complete structured data (sql tables and KG), as described in Motivation Section 2). Entity centric tables are semi-structured in nature and hold data in complex implicit form (see, for example, the discussion on semi-structured tables in Gupta et al, ACL 2020)
>
> Semi-structured data lies in a realm between raw, unstructured text and rigidly structured content. This data landscape, where structured frameworks interweave with free-form text, spans the gamut from extensive verbosity like web pages, to succinct instances such as fact sheets, information tables, and technical specifications. Unlike databases, this type of data isn't uniformly structured; it can be a heterogeneous assortment without preset schemas. Adding to the complexity, explanatory text that imparts context isn't always at hand. Nonetheless, we frequently deduce insights from such diverse and incomplete data, bridging information gaps based on our expectations about relationships within. Navigating semi-structured information necessitates a broad range of reasoning skills. We're tasked with comprehending a makeshift layout composed of elements like text snippets, form fields, or even substructured components like lists. Querying this data calls for various levels of inference, ranging from straightforward lookups (e.g. in athlete table of Petya in paper known born place), to lexical deductions (such as understanding that single  (WS) and double game (WD) format, junior championship vs senior events of badminton), to grasping the nature of content within cells (the structure of event, tournament name, tournament year, medals tally and tournament type). Additionally, we might find ourselves aggregating insights across multiple rows (such as understanding that dressage is a non-contact sport in which both genders compete), or even conducting intricate reasoning that melds temporal details with general knowledge.
>
> Moreover, in response to the reviewer's query, we will present statistics related to the table-specific datasets, and expand Appendix Table 31. In a nutshell, prior datasets primarily consist of non-temporal questions, with the rare temporal questions being explicit and not-grounded in commonsense and general world knowledge essential for real world temporal reasoning. It's worth noting that the scope of reasoning within these datasets is limited, focusing solely on explicit arithmetic temporal reasoning (SQL query oriented) applied to structured SQL-type tables, rather than the more intricate semi-structured infobox-style tables with commonsense and world knowledge.
>
> However, infoboxes are highly close to standard knowledge bases, particularly Wikidata (they are automatic conversions between Wikidata and Infoboxes, and Wikidata is generally much more complete).
>
> When comparing Wikidata to the Infobox style, distinct differences arise in how information is organized. Wikidata exhibits a more structured approach, akin to a knowledge graph.
>
> For instance, when considering birth details of a person, Wikidata neatly segregates information into "birth date," "birth place," and "birth name." Conversely, in semi-structured infoboxes, these details often amalgamate under a single term like "Born." Moreover, the manner in which relationships are presented demonstrates a divergence. In Wikidata, relationships are systematically delineated. For instance, instead of a broad "spouse" label, one might encounter separate entries for "husband" and "wife," offering clearer insights. In contrast, an infobox might encompass a single "spouse" entry without specifying the gender.
>
> Temporal details find distinct treatment as well. Wikidata distinctly separates "start date" and "end date," yielding precise timeline information. This stands in contrast to infoboxes, where these details could be condensed into single terms like "service," potentially necessitating further interpretation. Wikidata's penchant for hierarchy is evident in how complex terms are broken down. For instance, a "government official" could be subcategorized as "president," "prime minister," and more. In contrast, infoboxes might lack this hierarchical clarity, opting for more generalized terms.
>
> Granular attributes shine in Wikidata, with individual specifications for attributes like "awards," enabling a detailed breakdown of accolades. Conversely, infoboxes could consolidate these attributes, obscuring the specifics of received awards. When it comes to event descriptions, Wikidata adopts a distinction between "start time" and "end time," leading to lucid event elucidations. In infoboxes, these might be captured by a singular term, potentially devoid of temporal context.
>
> Lastly, Wikidata's categorization of properties imparts a structured approach to data. In contrast, infoboxes may not adhere to a similar systematic categorization, potentially leading to ambiguity. Collectively, these instances highlight the structured nature of Wikidata in contrast to the more succinct, semi-structured implicit knowledge of infoboxes.
>
>
> As infoboxes are highly structured (compared to Web tables), we could translate them to Wikidata and apply existing datasets and algorithms.
>
> Indeed, this approach holds promise, although it's worth noting that transforming them into a clean and fully structured Wikidata format is in itself a challenging task, as highlighted earlier. Nevertheless, it presents an interesting opportunity to explore the capabilities of state-of-the-art language models in achieving this conversion. However, it's important to acknowledge that tables not found on Wikipedia, such as those containing e-commerce attribute values, research grants, medical reports, financial company data, etc., pose their own challenges when it comes to transitioning them into structured knowledge formats like Wikidata.
>
> For the baselines, only raw language models are considered, whereas more advanced solutions could have been adopted (also briefly mentioned at the end). Although it is not the core goal of the paper, it could help future works.
>
> Certainly, while there's room for experimentation with more complex prompting approaches, it's essential to note that this falls outside the current scope and objectives of the paper. As a result, we've outlined this avenue for exploration in the section dedicated to future directions.
>
> W2: The authors do not explicitly evaluate the quality of the dataset. In section 3.3, they give good criteria for the evaluation (can we answer only with the table? Is the answer ambiguous? Is it an opinion?). Still, they are not used for the evaluation of the dataset. The human accuracy is about 86% for the exact match. Where do the errors come from (formulation? Hard questions?   = misunderstanding?)
>
> For the test and development sets, we uphold stringent quality control, manually scrutinizing each Table QA (with input from three experts) following the initial automated script-based filtering, all aimed at securing high-quality complex temporal questions. Additionally, we rectify answer units and address spelling errors during our quality filtering process. We prioritize questions that involve intricate temporal reasoning and exhibit an abstract nature. We filter questions where the answers are directly stated in either the question itself or the associated tables. We will mention these quality filtering steps briefly in Section 3.3, dataset validation, and in more detail in the appendix.
>
> In certain instances, we encountered questions that demanded external knowledge beyond common sense. These were selectively filtered out. The majority of errors were observed in questions that involvedd unusually complex reasoning. Additionally, many errors stemmed from minor miscalculations in date-time questions, typically off by a year. Ambiguities sometimes arose, such as when multiple answer forms hold the same meaning (e.g., "365 days," "12 months," and "1 year"). Furthermore, we make certain that our assessment script does not penalize either the model or the human verifier for any such unit conversion issues. Notably, only a small fraction of questions, namely less than 10% for the development set, less than 7% for the Head set, and less than 11% for the Tail set, were ambiguous and lacked a clear majority, as detailed in Table 8. To provide a broader perspective, we intend to include a cumulative count of majority agreements with respect to varying numbers of annotators in the final paper version.
>
> Furthermore, less than 3% of the questions displayed subjectivity or opinion-based characteristics during the test and development set filtering process. All these nuanced details will be comprehensively covered in the final version of the paper.
>
> W3: More detail on the baseline would be nice. How is the input of the model formatted? This is quite an important point due to the structure of infoboxes. Besides, I wish the authors would have tried to adapt state-of-the-art architecture for (temporal) question answering.
>
> We experimented with three prompts, each featuring detailed instructions similar to those given to human verifiers. These prompts were based on three distinct table representations using different delimiters. Our selection process involved choosing the prompt that yielded the best performance. We present the table input in a linear format, akin to the approach adopted in TabFact (Chen et al., 2020) and InfoTabS (Gupta et al., 2020). Here, we employ a distinctive denominator token to demarcate rows using “;” and columns using “:”. We explored alternative delimiters like “|” and “#” as well.
>
> We also experimented with an approach involving the conversion of tables into paragraphs before feeding them to the models. However, this method exhibited the unintended outcome of the models introducing extraneous external information. All these specific details will be elaborated upon in the appendix, along with additional information available on the public GitHub repository that we plan to release upon acceptance. Just to again emphasize that the performance variation was very marginal (< 1% in the  F1-score and 0.75 in the exact match).
>
> Many modern architectural models applied to tabular datasets take an extractive approach, which is generally characterized by non-generative encoder-only models. As an intentional choice, this work focuses on cutting-edge generative large language models, such as FlanT5 and GPT. Furthermore, several of these alternate models are either not publicly available or are proprietary in nature. TabT5, for example, falls into this category and is impossible to fine-tune, resulting in inferior performance datasets when subjected to zero or few-shot testing conditions.
>
> W4: Some analyses in section 5 are sometimes statistically irrelevant. I wish there were p-values. Some sample sizes are too small to draw conclusions (1, 16, 21, 22).
>
> Yes, we do agree that some of analysis on the test set have very few samples. We didn’t include this in the main paper due to page limits.  Therefore we plan to provide the analysis for the train set also in the appendix. We will also add p-values wherever possible for tables in section 5. Thanks for pointing this out.
>
> Other points:
>
> O0: The authors could have provided the dataset and code for the baselines on an anonymous link. It is currently impossible to check what the authors said.
>
> We will release the data and code with the final version. We put an anonymous link to follow the ACL anonymity deadline.
>
> O1: The paper's title is strangely formatted: TEMPTABQA is composed of several words, making it hard to copy-paste it (and parse for tools like Google Scholar).
>
> We will fix this and put something similar in the final version.
>
> O2: Would it be helpful to include synonym answers for a better exact match score? For example: "How many years..." could lead to "3", "3 years", "three," or "three years." Same thing for dates. Did you have a constraint that the Turks should or should not put the unit?
>
> You raise a valid point. As previously discussed, we've taken steps to ensure that our evaluation script doesn't unfairly penalize both human annotators and the model for providing answers in different formats. We've crafted regex rules that encompass a variety of such scenarios, and we've further validated this through human verification of numerous samples. Our annotators were instructed to provide succinct and essential information in their answers. While the majority of annotators followed the guidelines diligently, a few instances did deviate despite clear instructions. As part of our final dataset preparation, we plan to refine and include both the cleaned test version and the original test version to address these concerns.
>
> O3: Line 42-44: Research on the temporal aspect was limited.
>
> Thanks, we will fix this writing incoherency issues as pointed out by the reviewer.
> O4: Can a question have several correct answers?
> We request that annotators only write unambiguous questions. However, several format representations for the same answer are possible. For example, 1 year, 365 days, and 12 months can all be valid answers for the same question.
>
> O5: Section 5: cite the papers for the models you use.
>
> We will fix this writing issue, Thanks for pointing this out.
>
> O6: It is a pity the authors were not able to include bigger models for the finetuned models, like LLaMa or Bloom. I understand the technical difficulties, though. Algorithms like PEFT can help.
>
> We did not test PEFT, but we will see if we can fine-tune LLaMA on the dataset. Bloom is primarily bilingual; therefore, we didn't use it for our research. Of course, whether we can do so will depend on compute resource availability.
>
> O7: In the table, put the best results in bold and the second best underlined.
>
> Great suggestion; we would do that.
>
> O8: The abbreviations F1, EM, R1, R2, MET should be introduced somewhere.
>
> Thanks for pointing this out; we will add the full form.
>
> O9: Line 354: Is it surprising?
>
> Yes, the observation was not surprising, but the huge gap was a novel observation.
>
> O10: Paragraph line 366: The difference between head and tail is slim for finetuned models. Why? This should be reported.
>
> This phenomenon can be attributed primarily to the knowledge transfer occurring between less common and widely recognized sports tables. These tables share numerous common attributes and exhibit similarity in the types of questions posed. To underscore this observation, we intend to provide specific sport-related results in the appendix. Additionally, we plan to summarise these findings in the main experimental section for greater clarity.
>
> O11: Section 6 is not helpful. It is a kind of conclusion in the middle of the paper that only repeats the previous section. I do not see how the future directions are related to the previous discussions. I would put the section at the end.
>
> We would move the section 6 to the end of the paper.
>
> O12: What does HIT mean in Section D?
>
> A Human Intelligence Task, or HIT, is a question that needs an answer. A HIT represents a single, self-contained, virtual task that a Worker can work on, submit an answer, and collect a reward for completing. HITs are created by Requester customers in order to be completed by Worker customers.
>
> Questions For The Authors:
>
> Q1: Can you adapt state-of-the-art algorithms to your dataset? Which ones and how?
>
> We can adapt the existing generative encoder-decoder and decoder model such as TabT5, LLAMA, Falcon. We can also adapt existing KG based temporal QA model by first converting the table to KG’s. However this could be challenging as already discussed before.
>
> Q2: How is the input of the models formatted?
>
> We present the table input in a linear format, akin to the approach adopted in TabFact (Chen et al., 2020) and InfoTabS (Gupta et al., 2020). Here, we employ a distinctive denominator token to demarcate rows using “;” and columns using “:”. We explored alternative delimiters like “|” and “#” as well.
>
> We also experimented with an approach involving the conversion of tables into paragraphs before feeding them to the models. However, this method exhibited the unintended outcome of the models introducing extraneous external information. All these specific details will be elaborated upon in the appendix, along with additional information available on the public GitHub repository that we plan to release upon acceptance.
>
>
> Q3: More generally, answer the remarks above.
>
> Please see above

---

### Official Review · Reviewer_uQYi · 2023-08-04

**Soundness:** 5

**Excitement:**

3: Ambivalent: It has merits (e.g., it reports state-of-the-art results, the idea is nice), but there are key weaknesses (e.g., it describes incremental work), and it can significantly benefit from another round of revision. However, I won't object to accepting it if my co-reviewers champion it.

**Paper Topic And Main Contributions:**

This paper introduces the task of answering questions about temporal phenomena, with reference to a semi-structured table. This paper describes how a new dataset was built for this task, by collecting infoboxes from Wikipedia and having crowdworkers write questions about them. The new dataset is analyzed thoroughly. In addition, the paper contains results, thoroughly analyzed as well, with several state-of-the-art NLP models.

**Questions For The Authors:**

A. What is a "temporal question"? Since this is an important concept in the paper, it would be nice to have a definition early on.

B. How were GPT models prompted? Was there any randomness in the generation? This is important for reproducibility.

C. Line 497 Future Directions: This paragraph lists research that may be done, but hasn't been done. Why include that in a paper, and dedicate so much space to it?

**Reasons To Accept:**

- The paper shows good knowledge of related work
- The annotation process is sound and high-quality
- The paper contains a thorough statistical analysis of the new dataset
- The paper contains a thorough analysis of experimental results, which yields insights into some of the models

**Reasons To Reject:**

- The task that this paper addresses is not particularly exciting, and adjacent tasks have already been explored
- The paper spends considerable space on research that hasn't been done but may be done some day (starting on Line 497)

**Reproducibility:**

4: Could mostly reproduce the results, but there may be some variation because of sample variance or minor variations in their interpretation of the protocol or method.

**Reviewer Confidence:**

4: Quite sure. I tried to check the important points carefully. It's unlikely, though conceivable, that I missed something that should affect my ratings.

---

> ### Author Rebuttal · Authors · 2023-08-29
>
> Thank you for taking the time to read our article and provide valuable feedback.  We appreciate your thoughts and concerns and address them below:
>
> Q. The task that this paper addresses is not particularly exciting, and adjacent tasks have already been explored
>
> Complex temporal problems must be solved for a variety of crucial applications. It is essential for getting accurate historical insights when conducting historical studies or teaching. It provides financial analysts with historical financial data to help them make informed investment decisions in financial analysis. It grants access to clinical trial timetables and historical medical data for use in medical research. It is utilised by solicitors to study earlier court documents. Through the use of historical data, journalists may enhance their stories, linguists can analyse linguistic patterns, and corporations can improve their supply networks. Notably, this adaptable tool is useful to environmental researchers, policy analysts, travellers, software developers, and archivists. This clearly underlines its importance in gathering insightful information from structured tables spanning several areas. We'll proactively discuss these aspects in detail in the motivation section (Section 2) of the paper.
>
> Our study focuses on complex semi-structured data, notably entity-centric infobox tables. In contrast to earlier investigations that explored temporal reasoning in unstructured contexts such as paragraphs and knowledge graphs, our work focuses on these contexts in complex semi-structured data. This focus has particular difficulties, as described in the section on motivation (Section 2), but it is also very important, as was already mentioned. As indicated in the related study section (Section 7), it is important to note that available datasets for comparable tasks typically comprise straightforward and explicit temporal queries (see Table 31 in Appendix). The dataset statistics in Section 3.2, however, show that our focus is on answering challenging implicit queries. The tables' sophisticated, semi-structured design is what causes this complexity. For a deeper understanding, we intend to supplement the work with a complete comparison, comparable to Section 3.2, presenting the question statistics of various non-temporal tabular datasets.
>
> Q. The paper spends considerable space on research that hasn't been done but may be done some day (starting on Line 497)
> To emphasize the importance of our suggested datasets for upcoming research in the NLP community, we included a one-paragraph section with directions for future use of TempTabQA. According to earlier and recent ACL recommendations, this is a recommended practice from a research viewpoint.
>
> Questions For The Authors:
>
> A. What is a "temporal question"? Since this is an important concept in the paper, it would be nice to have a definition early on.
>
> When we refer to "temporal questions," we are talking about questions that require reasoning based on time-related facts. This can be either explicit, involving dates, years, times, etc., or implicit, using temporal terms like "rank," "before," "after," "predecessor," "successor," and so on. We'll elaborate on this definition along with examples in the introduction section (Section 1) of the paper.
>
> B. How were GPT models prompted? Was there any randomness in the generation? This is important for reproducibility.
>
> Indeed, there was a little randomness in the GPT-3.5 model, while the GPT-4 model displayed almost no randomness. However, variations in performance when altering prompts and regenerating content were very small. We experimented with three prompts, each featuring detailed instructions similar to those given to human verifiers. These prompts were based on three distinct table representations using different delimiters. Our selection process involved choosing the prompt that yielded the best performance. We will elaborate these details in the appendix, along with additional information available on the public GitHub repository that we plan to release upon acceptance. We wish to again emphasise that the performance variation was marginal (< 1% in F1-score and 0.75 in exact match).
>
> C. Line 497 Future Directions: This paragraph lists research that may be done, but hasn't been done. Why include that in a paper, and dedicate so much space to it?
>
> To emphasise the importance of our suggested datasets for upcoming research, in the NLP community, we included a one-paragraph section with directions for future use of TempTabQA. According to earlier and recent ACL recommendations, this is a recommended practice from a research viewpoint

---

### Official Review · Reviewer_tNdZ · 2023-08-14

**Soundness:** 4

**Excitement:**

4: Strong: This paper deepens the understanding of some phenomenon or lowers the barriers to an existing research direction.

**Missing References:**

Some works on temporal QA (both over text and KBs) could be referenced:

1.A Benchmark for Generalizable and Interpretable Temporal Question Answering Over Knowledge Bases”, in Arxiv, 2022.

2. Complex Temporal Question Answering on Knowledge Graphs. Zhen Jia, Soumajit Pramanik, Rishiraj Saha Roy, Gerhard Weikum

3. Targeted Extraction of Temporal Facts from Textual Resources for Improved Temporal Question Answering over Knowledge Bases. Nithish Kannen, Udit Sharma, Sumit Neelam, Dinesh Khandelwal, Shajith Ikbal, Hima Karanam and L. Venkata Subramaniam, “

**Paper Topic And Main Contributions:**

The paper introduces a new task of temporal question answering over semi-structured tables. For the same, they introduce a new dataset which comprises 11,454 question-answer pairs extracted from 1,208 Wikipedia Infobox tables spanning over 90 distinct domains. They experiment with pre-trained language models on the dataset and report that they lag behind human performance. They provide a detailed analysis breakdown of model performance against humans and additionally highlight the future research directions to guide the community.

**Questions For The Authors:**

1)The paper could benefit if the authors highlight the real world implications of such a task. "Can current systems reason about such information in semi-structured tables?" why is answering this important? What systems can be built using this?

2) Were all the annotated train samples used? Or was there a quality-filtering step before that?

**Reasons To Accept:**

1) They introduce a new task and accompany that with a sufficiently large dataset. The new task could be of interest to the NLP community.

2) The paper is well structured and clear. It is easy to read and has sufficient details about the dataset

3) The requirement for such a dataset is well motivated, highlighting the gaps in the existing literature and datasets.

4) A well written section about future directions is encouraging.

5) Detailed breakdown of performance to highlight where exactly the models lack in comparison to humans

**Reasons To Reject:**

1) The experimental section lacks complex modelling techniques. The authors have simply used existing LLMs and modelling techniques to compare performance. A novel modelling technique would strengthen the paper.

2) The current manuscript only introduces a new dataset with minimal contributions on the modelling or science part. The authors claim that existing modelling approaches are insufficient, however they fail to propose even simple improvements to alleviate some of the shortcomings. I am afraid there isn't enough novelty or scientific contributions in this work to qualify as a research paper at a top-tier conference.

3) Some unnecessary details can be moved to the Appendix. For example, the prompts and instructions used to guide the annotator may not be important to the reader. More focus is needed on the novelty.

4) Nitpick: The main sections of the dataset analysis contains very minimal pictorial representation. A lot of tables with numbers makes it hard to read. A few figures and bar graphs could be some good choices to convey the same.

**Reproducibility:**

4: Could mostly reproduce the results, but there may be some variation because of sample variance or minor variations in their interpretation of the protocol or method.

**Reviewer Confidence:**

5: Positive that my evaluation is correct. I read the paper very carefully and I am very familiar with related work.

**Typos Grammar Style And Presentation Improvements:**

Some figures could be used for representing the data composition. No other major concerns.

---

> ### Author Rebuttal · Authors · 2023-08-29
>
> Thank you for taking the time to read our article and provide valuable feedback.  We appreciate your thoughts and concerns and address them below:
>
> Reasons to Reject
>
> Q1. The experimental section lacks complex modelling techniques. The authors have simply used existing LLMs and modelling techniques to compare performance. A novel modelling technique would strengthen the paper.
>
> The TempTabQA dataset primarily consists of abstractive reasoning questions. Therefore, an ideal model for this task should possess generative capabilities. Existing datasets for tabular questions have predominantly featured extractive queries (See section 2 Motivation), resulting in the prevalence of encoder-focused models for diversity. In our study, we pivot towards state-of-the-art generative models, commonly referred to as Language Model (LLM) variants, encompassing both open-source options like FlanT5, BART, and T5, as well as proprietary alternatives like GPT-3.5 and GPT-4.
>
> At a high level, we agree with the reviewer that a novel modelling technique such as one involving temporal graphs (e.g. EXAQT, Jia et. al, 2020),  would be a great direction for further work.
>
> Other open-source models either remained unavailable or hadn't gained widespread usage by the time of our paper's submission. It's worth noting that models such as the TabT5 couldn't be included due to proprietary industry restrictions (not open sourced). The primary objective of our paper was to shed light on the temporal question-answering challenge within a tabular context. Our contribution lies in presenting a robust foundational problem and strong baselines for addressing this issue. Moving forward, it would be intriguing to explore alternative models and delve into intricate prompting techniques, as outlined in the potential avenues for future research (as discussed in Section 6).
>
> Q2. The current manuscript only introduces a new dataset with minimal contributions on the modeling or science part. The authors claim that existing modelling approaches are insufficient, however they fail to propose even simple improvements to alleviate some of the shortcomings. I am afraid there isn't enough novelty or scientific contributions in this work to qualify as a research paper at a top-tier conference.
>
> We respectfully disagree with the assertion that our work lacks novelty or scientific contribution. Quite the contrary, our paper pioneers the creation of complex temporal question answering datasets. Furthermore, it introduces the first complex question answering a dataset tailored to entity-centric tables. In contrast, prior endeavours such as "Learning to Answer Questions from Wikipedia Infoboxes" by Morales et al. (2016) (will cite in the introduction section 1) lack the complexity of our questions, and their dataset remains proprietary. Notably, our paper incorporates robust state-of-the-art baselines, leveraging contemporary Language Model Models (LLMs) and revealing their comparative shortcomings in comprehending and reasoning over the TempTabQA dataset.
>
> As previously mentioned, the primary objective of our paper was to introduce an innovative challenge: tackling intricate temporal questions within the context of entity-centric tables. The TempTabQA dataset demands sophisticated reasoning, incorporating temporal common sense, and adeptly managing arithmetic and numerical aspects. Our work serves to elucidate the distinct temporal specificity of the proposed dataset, setting it apart from existing models. This differentiation and the multitude of temporal reasoning challenges it poses are extensively illuminated in Section 3.2 through a comprehensive array of statistics and analyses. In line with the forward-looking orientation of our work, as outlined in the "Future Directions" section (section 6), we believe this study opens up numerous pathways for exploring intricate temporal reasoning on semi-structured tables and other complex structures.
>
> Q3. Some unnecessary details can be moved to the Appendix. For example, the prompts and instructions used to guide the annotator may not be important to the reader. More focus is needed on the novelty.
>
> Thanks for the suggestion, we will provide the annotation details to the appendix, thereby streamlining the core content for enhanced readability. Moreover, we recognize the need to reinforce the focus on the novelty of our work. To address this, we will strengthen the motivation surrounding the application of complex temporal reasoning within the domain of entity-centric tables. Doing so will also highlight the significance of the problem at hand and provide a stronger foundation for the originality of our contributions.
>
> Q4. Nitpick: The main sections of the dataset analysis contains very minimal pictorial representation. A lot of tables with numbers makes it hard to read. A few figures and bar graphs could be some good choices to convey the same.
>
> As recommended by the reviewer, we will shift some results from tabular representation to pictorial form, especially line and bar charts. Thanks for pointing out these presentation issues.
>
> Questions to Authors
>
> Q1. The paper could benefit if the authors highlighted the real world implications of such a task. "Can current systems reason about such information in semi-structured tables?" Why is answering this important? What systems can be built using this?
>
> A1. Complex temporal question answering applied to entity-centric semi-structured tables, like Wikipedia infoboxes, has broad applicability across various fields. In historical research and education, it helps scholars, historians, and students extract precise historical details, while in financial analysis, it empowers analysts with historical financial data for informed investment decisions. In medical research, it aids in accessing historical medical data and clinical trial timelines, and legal professionals use it to review historical legal records. Journalists gain historical context, linguists analyze language dynamics, and businesses optimize supply chains through historical data. Environmental researchers, policy analysts, travelers, software developers, and archivists all benefit from this versatile tool. This underscores its significance in extracting valuable insights from structured tables across diverse domains. We will mention these in the motivation section 2 of the paper in detail.
>
> Q2. Were all the annotated train samples used? Or was there a quality-filtering step before that?
>
> A2. We employ simple scripts to eliminate non-temporal and basic extractive questions from the training set before fine-tuning. For the test and development sets, we uphold stringent quality control, manually scrutinizing each Table QA (with input from three experts) following the initial automated script-based filtering, all aimed at securing high-quality complex temporal questions. Additionally, we rectify answer units and address spelling errors during our quality filtering process. We prioritize questions that involve intricate temporal reasoning and exhibit an abstract nature. We filter questions where the answers are directly stated in either the question itself or the associated tables. We will mention these quality filtering steps briefly in Section 3.3, Dataset Validation, and in more detail in the appendix.
>
> Q3. Some works on temporal QA (both over text and KBs) could be referenced:
>
> A Benchmark for Generalizable and Interpretable Temporal Question Answering Over Knowledge Bases”, in Arxiv, 2022.
> Complex Temporal Question Answering on Knowledge Graphs. Zhen Jia, Soumajit Pramanik, Rishiraj Saha Roy, and Gerhard Weikum
> Targeted Extraction of Temporal Facts from Textual Resources for Improved Temporal Question Answering over Knowledge Bases. Nithish Kannen, Udit Sharma, Sumit Neelam, Dinesh Khandelwal, Shajith Ikbal, Hima Karanam and L. Venkata Subramaniam
>
> Missing References: Thanks for pointing out the missing references, we will add the mentioned paper in the related work.
>
> Q4. Typos Grammar Style And Presentation Improvements: Some figures could be used for representing the data composition. No other major concerns.
>
> As recommended by the reviewer, we will shift some results from tabular representation to pictorial form, especially line and bar charts. Thanks for pointing out these presentation issues.

---

### Meta-Review · Area_Chair_KmGp · 2023-09-10

**Recommendation:** 5

**Metareview:**

This paper introduces an innovative temporal question-answering task along with a dataset designed for semi-structured tables (info-boxes), which is a well-justified endeavor. The authors' commitment to elaborate more about the motivation behind this task, in response to reviewers' suggestions, is noteworthy. The paper's clarity in presentation and the comprehensive explanation of the data collection process are also notable strengths. Furthermore, the thorough and easily understandable future work section greatly enhances the paper's significance and its contribution to the research community.

I believe that the paper is of high quality. However, I think that if the authors had suggested some modeling modifications to existing methods in order to solve the task, it would have been a great nominee for the best paper award.

---

### Decision · Program_Chairs · 2023-10-07

**Decision:**

Accept-Main

**Comment:**

This paper introduces an innovative temporal question-answering task along with a dataset designed for semi-structured tables (info-boxes), which is a well-justified endeavor. The authors' commitment to elaborate more about the motivation behind this task, in response to reviewers' suggestions, is noteworthy. The paper's clarity in presentation and the comprehensive explanation of the data collection process are also notable strengths. Furthermore, the thorough and easily understandable future work section greatly enhances the paper's significance and its contribution to the research community.

I believe that the paper is of high quality. However, I think that if the authors had suggested some modeling modifications to existing methods in order to solve the task, it would have been a great nominee for the best paper award.